

**Changing pattern of ice flow and mass balance for glaciers discharging into the Larsen A and**
**B embayments, Antarctic Peninsula, 2011 to 2016**
Helmut Rott[1,2][*], Wael Abdel Jaber[3], Jan Wuite[1], Stefan Scheiblauer[1], Dana Floricioiu[3], Jan
Melchior van Wessem[4], Thomas Nagler[1], Nuno Miranda[5], Michiel R. van den Broeke[4]
[1] ENVEO IT GmbH, Innsbruck, Austria
[2] Institute of Atmospheric and Cryospheric Sciences, University of Innsbruck, Innsbruck, Austria
[3] Institute for Remote Sensing Technology, German Aerospace Center, Oberpfaffenhofen,
Germany
[4] Institute for Marine and Atmospheric Research, Utrecht University, Utrecht, the Netherlands
[5] European Space Agency/ESRIN, Frascati, Italy
*Correspondence to: Helmut.Rott@enveo.at





**Abstract**
We analyzed volume change and mass balance of outlet glaciers on the northern Antarctic Peninsula
over the periods 2011 to 2013 and 2013 to 2016, using high resolution topographic data of the
bistatic interferometric radar satellite mission TanDEM-X. Complementary to the geodetic method
applying DEM differencing, we computed the net mass balance of the main outlet glaciers by the
input/output method, accounting for the difference between the surface mass balance (SMB) and the
discharge of ice into an ocean or ice shelf. The SMB values are based on output of the regional
climate model RACMO Version 2.3p2. For studying glacier flow and retrieving ice discharge we
generated time series of ice velocity from data of different satellite radar sensor, with radar images
of the satellites TerraSAR-X and TanDEM-X as main source. The study area comprises tributaries
to the Larsen-A, Larsen Inlet, and Prince-Gustav-Channel embayments (region A), the glaciers
calving into Larsen B embayment (region B), and the glaciers draining into the remnant part of
Larsen B ice shelf in SCAR Inlet (region C). The glaciers of region A, where the buttressing ice
shelf disintegrated in 1995, and of region B (ice shelf break-up in 2002) show continuing losses in
ice mass, with significant reduction of losses after 2013. The mass balance numbers for grounded
glacier area of the region A are $B_n$ = -3.98 ± 0.33 Gt $a^{-1}$ during 2011 to 2013 and $B_n$ = -2.38 ± 0.18
Gt $a^{-1}$ during 2013 to 2016. The corresponding numbers for region B are $B_n$ = -5.75 ± 0.45 Gt $a^{-1}$
and $B_n$ = -2.32 ± 0.25 Gt $a^{-1}$. The mass losses in region C during the two periods were modest, $B_n$ =
-0.54 ± 0.38 Gt $a^{-1}$, respectively $B_n$ = -0.58 ± 0.25 Gt $a^{-1}$. The main share in the overall mass losses
of the region were contributed by two glaciers: Drygalski Glacier contributing 61 % to the mass
deficit of region A, and Hektoria and Green glaciers accounting for 67 % to the mass deficit of
region B. Hektoria and Green glaciers accelerated significantly in 2010/2011, triggering elevation
losses up to 19.5 m $a^{-1}$ on the lower terminus and a rate of mass depletion of 3.88 Gt $a^{-1}$ during the
period 2011 to 2013. Slowdown of calving velocities and reduced calving fluxes in 2013 to 2016
coincided with years when the sea ice cover in front of the glaciers persisted during summer.






## 1. Introduction

The disintegration of the ice shelves in Prince-Gustav-Channel and the Larsen A embayment in January 1995 (Rott et al., 1996) and the break-up of the northern and central sections of Larsen B embayment in March 2002 (Rack and Rott, 2004; Glasser and Scambos, 2008) triggered near-immediate acceleration of the outlet glaciers previously feeding the ice shelves, resulting in major mass losses due to increased ice discharge (Rott et al., 2002; De Angelis and Skvarca, 2003; Scambos et al., 2004; Scambos et al, 2011). Precise, spatially detailed data on flow dynamics and mass balance of these glaciers since ice-shelf disintegration are essential for understanding the complex glacier response to the loss of ice shelf buttressing, as well as to learn about processes controlling the adaptation to new boundary conditions. Furthermore, due to the complex topography of this region, spatially detailed data on glacier surface elevation change and mass balance are the key for reducing the uncertainty of northern Antarctic Peninsula (API) contributions to sea level rise.

Several studies dealt with mass balance, acceleration and thinning of glaciers after disintegration of the Larsen A and B ice shelves, with the majority focusing on glaciers of the Larsen B embayment. A complete, detailed analysis of changes in ice mass was performed by Scambos et al. (2014) for 33 glacier basins covering the API mainland and adjoining islands north of 66°S, using a combination of digital elevation model (DEM) differencing from optical stereo satellite images and repeat-track laser altimetry from the Ice, Cloud, and Land Elevation Satellite (ICESat). The DEM difference pairs cover the periods 2001-2006, 2003-2008, and 2004-2010 for different sections of the study area, and are integrated with ICESat data of the years 2003 to 2008. A detailed analysis of surface elevation change and mass depletion for API outlet glaciers draining into the Larsen-A, Larsen Inlet, and Prince-Gustav-Channel (PGC) embayments during 2011 to 2013 was reported by Rott et al. (2014), based on topographic data of the TanDEM-X/TerraSAR-X satellite formation. With an annual loss in ice mass of $4.21 \pm 0.37$ Gt $a^{-1}$ during 2011-2013 these glaciers were still largely out of balance, although the loss rate during this period was diminished by 27% compared to the loss rate reported by Scambos et al. (2014) for 2001 to 2008. Studies on frontal retreat, ice velocities, and ice discharge, based on remote sensing data of the period 1992 to 2014, are reported by Seehaus et al. (2015) for the Dinsmoor–Bombardier–Edgeworth glacier system previously feeding the Larsen A ice shelf and by Seehaus et al. (2016) for glaciers of Sjögren Inlet previously feeding the PGC ice shelf.

As observed previously for Larsen A (Rott et al., 2002), the major outlet glaciers to the Larsen B embayment started to accelerate and get thinner immediately after the collapse of the ice shelf (Rignot et al., 2004; Scambos et al., 2004; De Rydt et al., 2015). The patterns of acceleration,



thinning and change of frontal position have been variable in time and space. After strong
acceleration during the first years, some of the main glaciers slowed down significantly after 2007,
resulting in major decrease of calving fluxes. Other glaciers continued to show widespread
fluctuations in velocity, with periods of major frontal retreat alternating with stationary positions or
intermittent frontal advance (Wuite et al., 2015). The remnant section of Larsen B ice shelf in
SCAR inlet started to accelerate soon after the central and northern sections of the ice shelf broke
away, triggering modest acceleration of the main glaciers flowing into the SCAR inlet ice shelf
(Wuite et al., 2015; Khazendar et al., 2015).
Several publications reported on ice export and mass balance of Larsen-B glaciers. Shuman et al.
(2011) derived surface elevation change from optical stereo satellite imagery and laser altimetry of
ICESat and the airborne Airborne Topographic Mapper (ATM) of NASA's IceBridge program. For
the period 2001 to 2006 they report a combined rate of mass losses of 8.4 ± 1.7 Gt a$^{-1}$ for the
glaciers discharging into Larsen B embayment and SCAR Inlet, excluding ice lost by frontal retreat.
ICESat and ATM altimetry measurements spanning 2002–2009 show for lower Crane Glacier a
period of very rapid drawdown between September 2004 and September 2005, bounded by periods
of more moderate rates of surface lowering (Scambos et al., 2011). Rott et al. (2011) derived
velocities and ice discharge of the nine main Larsen B glaciers in pre-collapse state (1995 and 1999)
and for 2008-2009, estimating the mass imbalance of these glaciers in 2008 at 4.34 ± 1.64 Gt a$^{-1}$.
Berthier et al. (2012) report a mass loss rate of 9.04 ± 2.01 Gt a$^{-1}$ for Larsen B glaciers, excluding
SCAR inlet, for the period 2006 to 2010/2011, based on altimetry and optical stereo imagery.
Scambos et al. (2014) analysed changes in ice mass from ICESat data spanning September 2003 to
March 2008 and stereo image DEMs spanning 2001/2002 to 2006. They report a combined rate of
mass losses of 7.9 Gt a$^{-1}$ for the tributaries of the Larsen B embayment and of 1.4 Gt a$^{-1}$ for the
tributaries to SCAR Inlet ice shelf. Wuite et al. (2015) report for main outlet glaciers strongly
reduced calving fluxes during the period 2010 to 2013 compared to the first few years after ice shelf
collapse.
We use high resolution data of surface topography derived from synthetic aperture radar
interferometry (InSAR) satellite measurements for retrieving changes in glacier volume and
estimating glacier mass balance over well-defined epochs for API outlet glaciers along the Weddell
Coast between PGC and Jason Peninsula. In addition, we generate ice velocity maps to study the
temporal evolution of ice motion and derive the ice discharge for the major glacier drainage basins.
We compute the mass balance also by means of the input-output method (IOM), quantifying the
difference between glacier surface mass balance (SMB) and the discharge of ice into the ocean or
across the grounding line to an ice shelf. The SMB estimates are obtained from output of the



regional atmospheric climate model RACMO Version 2.3p2 at grid size of ∼ 5.5 km (van Wessem
et al., 2016; 2017).
Volume change and mass balance of glaciers discharging into the PGC, Larsen Inlet and Larsen A
embayments were derived by Rott et al. (2014) for the period 2011 to 2013, applying TanDEM-X
DEM differencing. Here we extend the observation period for the same glacier basins by covering
the time span 2013 to 2016. Furthermore, we present time series of surface velocity starting in
1993/1995 in order to relate the recent flow behavior to pre-collapse conditions.
For glaciers of the Larsen-B embayment we generated maps of surface elevation change by
TanDEM-X DEM differencing for the periods 2011 to 2013 and 2013 to 2016. From these maps we
derived mass changes at the scale of individual glacier drainage basins. In addition, we obtained
mass balance estimates for the eight main glaciers by the input/output method and compare the
results of the two independent methods. A detailed analysis of surface velocities of Larsen B
glaciers for the period 1995 to 2013 was presented by Wuite et al. (2015). We extend the time series
to cover glacier velocities up to 2016.
These data sets disclose large temporal and spatial variability in ice flow and surface elevation
change between different glacier basins and show ongoing loss of grounded ice. This provides a
valuable basis for studying factors responsible for instability and downwasting of glaciers and for
exploring possible mechanisms of adaptation to new boundary conditions.
**2. Data and methods**
**2.1 DEM differencing using TanDEM-X interferometric SAR data**
The study is based on remote sensing data from various satellite missions. We applied DEM
differencing using interferometric SAR data (InSAR) of the TanDEM-X mission to map the surface
elevation change and retrieve the mass balance for 24 catchments on the API east coast between
PGC and Jason Peninsula (Supplement, Table S1). Large glaciers are retained as single catchments
whereas smaller glaciers and glaciers that used to share the same outlet are grouped together. For
separation of glacier drainage basins inland of the frontal areas the glacier outlines of the
Glaciology Group, University of Swansea, are used which are available at the GLIMS data base
(Cook et al., 2014). We updated the glacier fronts for several dates of the study period using
TerraSAR-X, TanDEM-X and Landsat-8 images. Catchment outlines and frontal positions in 2011,
2013 and 2016 are plotted in a Landsat image of 2016-10-29 (Supplement, Figures S1 and S2).
The TanDEM-X mission (TDM) employs a bi-static interferometric configuration of the two
satellites TerraSAR-X and TanDEM-X flying in close formation (Krieger et al., 2013). The two
satellites form together a single-pass synthetic aperture radar (SAR) interferometer, enabling the





acquisition of highly accurate cross-track interferograms that are not affected by temporal
decorrelation and variations in atmospheric phase delay. The main objective of the mission is the
acquisition of a global DEM with high accuracy. The 90 % relative point-to-point height accuracy
for moderate terrain is ±2 m at 12 m posting (Rossi et al., 2012; Rizzoli et al., 2012). Higher relative
vertical accuracy can be achieved for measuring elevation change over time.
Our analysis of elevation change is based on DEMs derived from interferograms acquired by the
TanDEM-X mission in mid-2011, -2013 and -2016. SAR data takes from descending satellite orbits,
acquired in 2013 and 2016, cover the API east coast glaciers between 64° S and the Jason
Peninsula, as well as parts of the west coast glaciers (Supplement, Figure S3). For 2011 we
processed data takes covering the Larsen B glaciers. Over the Larsen A glaciers TDM data from
2011 and 2013 had been processed in an earlier study to derive surface elevation change (SEC). The
mid-beam incidence angle of the various tracks varies between 36.1 and 45.6 degrees. The height of
ambiguity (HoA, the elevation difference corresponding to a phase cycle of $2\pi$) varies between 20.6
m and 68.9 m, providing good sensitivity to elevation (Rott, 2009) (Supplement, Table S2). Only
track A has larger HoA and thus less height sensitivity; this track extends along the west coast and
covers only a very small section of study glaciers along the Weddell Coast.
We used the operational Integrated TanDEM-X Processor (ITP) of the German Aerospace Center
(DLR) to process the raw bistatic SAR data of the individual tracks into so-called Raw DEMs
(Rossi et al., 2012; Abdel Jaber et al., 2016). In the production line for the global DEM, which also
uses the ITP Processor, Raw DEMs are intermediate products before DEM mosaicking. An option
recently added to the ITP foresees the use of reference DEMs to support Raw DEM processing
(Lachaise and Fritz, 2016). We applied this option for generating the Raw DEMs, subtracting the
phase of the simulated reference DEM from the interferometric phase of the corresponding scene.
The recently released TanDEM-X global DEM with a posting of 0.4 arcsec was used as the main
source for the reference DEM. Although the relative elevation in output is not related to the
reference DEM, the presence of inconsistencies in the reference DEM may lead to artefacts in the
output DEM. Therefore some preparatory editing was performed: unreliable values were removed
based on the provided consistency mask of the global DEM and visual analysis and were substituted
by data of the Antarctic Peninsula DEM of Cook et al. (2012). The phase difference image, which
has a much lower fringe frequency, is unwrapped and summed up with the simulated phase image.
This option provides a robust phase unwrapping performance for compiling the individual DEMs.
By subtracting the two DEMs and accounting for the appropriate time span we obtain a surface
elevation rate of change map, with horizontal posting at about 12 m x 12 m.
For estimating the uncertainty of the TanDEM SEC maps we use a fully independent data set



acquired during NASA IceBridge campaigns that became available after the production of the TDM
SEC maps had been completed. Surface elevation rate of change data (dh/dt, product code
IDHDT4) derived from Airborne Topographic Mapper (ATM) swathes, acquired on 2011-11-14 and
2016-11-10, cover longitudinal profiles on six of our study glaciers (Studinger, 2014, updated
2017). Each IDHDT4 data record corresponds to an area where two ATM lidar swathes have co-
located measurements. The IDHDT4 data are provided as discrete points representing 250 m x 250
m surface area and are posted at about 80 m along-track spacing. We compare mean values of cells
comprising 7 x 7 TDM dh/dt pixels (12 m x 12 m pixel size) with the corresponding IDHDT4
points. Even though the start and end dates of the TDM and ATM data sets differ by a few months,
the agreement in dh/dt is very good. The root mean square differences (RMSD) of the data points
range from 0.14 m a$^{-1}$ to 0.35 m a$^{-1}$ for the different glaciers, and the mean difference of the ATM –
TDM data sets is dh/dt = -0.08 m a$^{-1}$. For the error analysis we assume that the differences result
from uncertainties in both data sets. The resulting RMSE for the TDM dh/dt cells is 0.20 m a$^{-1}$ over
the five year time span, and 0.39 m a$^{-1}$ and 0.58 m a$^{-1}$ for the three and two year time span,
respectively.
The agreement between the lidar and radar dh/dt data indicates that radar penetration is not an issue
for deriving elevation change from the SAR based DEMs of this study. This can be attributed to the
close agreement of the view angles in the corresponding SAR repeat data, acquired from the same
orbit track and beam, and to the consistency of radar propagation properties in the snow and firn
bodies. The latter point follows from the similarity of the backscatter coefficients of the
corresponding scenes, with differences between the two dates staying below 1 dB. The radar
backscatter coefficient can be used as indicator on stability in the structure and radar propagation
properties of a snow/ice medium which determine the signal penetration and the offset of the
scattering phase centre versus the surface (Rizzoli et al., 2017). The TSX and TDM SAR
backscatter images have high radiometric accuracy (absolute radiometric accuracy 0.7 dB, relative
radiometric accuracy 0.3 dB), well suitable for quantifying temporal changes in backscatter
(Schwerdt et al., 2010; Walter Antony et al., 2016).
The main outlet glaciers of the study area arise from the plateaus along the central API ice divide.
The plateaus stretch across elevations between about 1500 and 2000 m a.s.l. A steep escarpment,
dropping about 500 m in elevation, separates the plateau from the individual glacier streams and
cirques. The high resolution SEC maps, shown in Figures 1, 5, and 6, cover the areas below the
escarpment excluding parts of the steep rock- and ice- covered slopes along the glacier streams.
These gaps are due to the particular SAR observation geometry, with slopes facing towards the
illuminating radar beam appearing compressed (foreshortening) or being affected by superposition





of dual or multiple radar signals (layover) (Rott, 2009). On areas with gentle topography and on
slopes facing away from the radar beam (back-slopes) the surface elevation and its change can be
derived from the interferometric SAR images. In order to fill the gaps in areas of foreshortening and
layover, we checked topographic change on back-slopes. The TDM data set includes SEC data for
38 individual sections on back-slopes with mean slope angles ≥20 degrees, covering a total area of
787 km$^2$. The mean dh/dt value of these slopes is -0.054 m a$^{-1}$. The satellite derived velocity maps
show surface velocities <0.02 m d$^{-1}$ anywhere on the slope areas, indicating that dynamic effects are
insignificant for mass turnover. This explains the observed stability of surface topography.
There are some gaps in the SEC maps also on the plateau above the escarpment. The TDM SEC
analysis covers substantial parts (all together 2013 km$^2$) of the ice plateaus between 1500 m and
2000 m, the mean value dh/dt is -0.012 m a$^{-1}$. No distinct spatial pattern is evident. Considering the
small change of surface elevation in the available data samples of the ice plateau and on the slopes,
we assume stationary conditions for the unsurveyed slopes and the central ice plateau. For
estimation of uncertainty we assume for these areas a bulk uncertainty dh/dt = ± 0.10 m a$^{-1}$ for the
error budget of elevation change derived from DEMs spanning three years and dh/dt = ± 0.15 m a$^{-1}$
for DEMs spanning two years.
**2.2 Ice velocity maps and calving fluxes**
We generated maps of glacier surface velocity for several dates of the study period from radar
satellite images, extending the available velocity time series up to 2016. The main data base for the
recent velocity maps are repeat-pass SAR images of the satellites TerraSAR-X and TanDEM-X.
Gaps in these maps, primarily in the slowly moving interior, are filled with velocities derived from
SAR images of Sentinel-1 (S1) and of the Phased Array L-band SAR (PALSAR) on ALOS. We
applied offset tracking for deriving two-dimensional surface displacements in radar geometry and
projected these onto the glaciers surfaces defined by the ASTER-based Antarctic Peninsula digital
elevation model (API-DEM) of Cook et al. (2012). The velocity data set comprises the three
components of the surface velocity vector in Antarctic polar stereographic projection resampled to a
50 m grid.
The TerraSAR-X/TanDEM-X velocity maps are based on SAR strip map mode images of 11-day
repeat-pass orbits, using data spanning one or two repeat cycles. Due to the high spatial resolution
of the images (3.3 m along the flight track and 1.2 m in radar line-of-sight) velocity gradients are
well resolved. Regarding S1 we use single look complex (SLC) Level 1 products acquired in
Interferometric Wide (IW) swath mode, with nominal spatial resolution 20 m x 5 m (Torres et al.
2012; Nagler et al., 2015). Images of the Sentinel-1A satellite at 12-day repeat cycle cover the study



region since December 2014. Since September 2016 the area is also covered by the Sentinel-1B
satellite, providing a combined S1 data set with 6-day repeat coverage.
Wuite et al. (2015) estimate the uncertainty of velocity magnitude derived from TerraSAR-X 11-day
repeat pass images at $\pm$ 0.05 m d$^{-1}$. In order to check the impact of combining different ice velocity
products, we compared TerraSAR-X/TanDEM-X velocity maps of the study area, resampled to 200
m, with S1 velocity maps using data sets with a maximum time difference of 10 days. The overall
mean bias (S1 – TerraSAR-X/TanDEM-X) between the two data sets (sample 570,000 points) is
0.011 m d$^{-1}$ for velocity component Ve (easting) and -0.002 m d$^{-1}$ for Vn (northing), the RMSD is
0.175 m d$^{-1}$ for Ve and 0.207 m d$^{-1}$ for Vn. The good agreement of the mean velocity values points
out that velocity data from the two missions can be well merged.
In addition to the recently generated velocity products we use velocity data from earlier years for
the scientific interpretation which were derived from SAR data of various satellite missions,
including ERS-1, ERS-2, Envisat ASAR, and ALOS PALSAR (Rott et al., 2002; 2011; 2014; Wuite
et al., 2015).
In order to obtain mass balance estimates by the input/output method, we compute the mass flux F
across a gate of width Y [m] at the calving front or grounding line according to:

$$F_Y = \rho_i \int_0^Y [u_m(y)H(y)]\, dy$$

$\rho_i$ is the density of ice, $u_m$ is the mean velocity of the vertical ice column perpendicular to the gate,
and H is the ice thickness. We use ice density of 900 kg m$^{-3}$ to convert ice volume into mass. For
calving glaciers full sliding is assumed across calving fronts, so that $u_m$ corresponds to the surface
velocity, $u_s$, obtained from satellite data. For glaciers discharging into the SCAR Inlet ice shelf we
estimated the ice deformation at the flux gates applying the laminar flow approximation (Paterson,
1994). The resulting vertically averaged velocity for these glaciers is $u_m = 0.95\ u_s$. The ice thickness
at the flux gates is obtained from various sources. For some glaciers sounding data on ice thickness
are available, measured either by in situ or airborne radar sounders (Farinotti et al., 2013; 2014;
Leuschen et al. 2010, updated 2016). For glaciers with floating terminus the ice thickness is
deduced from the height above sea level applying the flotation criterion. For uncertainty estimates
of mass fluxes we assume $\pm$ 10 % error for the cross section area of glaciers with GPR data across
or close to the gates and $\pm$ 15 % for glaciers where the ice thickness is deduced from frontal height
above flotation. For velocities across the gates we assume $\pm$ 5 % uncertainty. For the uncertainty of
surface mass balance at basin scale, based on RACMO output available at monthly time scale, we
assume $\pm$ 15 % uncertainty.



**3. Elevation change and mass balance of glaciers north of Seal Nunataks**

**3.1 Elevation change and mass balance by DEM differencing**

The map of surface elevation change dh/dt from June/July 2013 to July/August 2016 for the glacier basins discharging into PGC, Larsen Inlet and Larsen A embayment is shown in Figure 1. The numbers on elevation change, volume change and mass balance, excluding floating glacier areas, are specified in Table 1. As explained in Section 2.1, for areas not displayed in this map (steep radar fore-slopes and the ice plateau above the escarpment) the available data indicate minimal changes in surface elevation so that stable surface topography is assumed for estimating the net mass balance.

For glaciers with major sections of floating ice and frontal advance or retreat the extent, SEC and volume change (including the subaqueous part) of the floating area and the advance/retreat area and volume are specified in Table 2. The area extent of floating ice is inferred from the reduced rate of SEC compared to grounded ice, using the height above sea level as additional constraint. Dinsmoor-Bombardier-Edgeworth glaciers (DBE, basin A4) had the largest floating area (56.2 km$^2$) extending about 8 km into a narrow fjord and showed also the largest frontal advance (11.7 km$^2$) between 2013 and 2016.

The mass depletion of grounded ice in the basins A1 to A7 (2.38 Gt a$^{-1}$) during the period 2013 to 2016 amounts to 60 % of the 2011 to 2013 depletion rate (3.98 Gt a$^{-1}$ for the grounded areas; Rott et al., 2014). The mass deficit is dominated by Drygalski Glacier (1.72 Gt a$^{-1}$ for 2013 to 2016), down from 2.18 Gt a$^{-1}$ for 2011 to 2013. A decline of mass losses between the first and second period is observed for all basins except A3 (Albone, Pyke, Polaris, Eliason glaciers, APPE) in Larsen Inlet which was approximately in balanced state during 2011 to 2016.

The altitude dependence of elevation change (dh/dt) for the three basins with the largest mass deficit is shown in Figure 2. Positive values in the lowest elevation zone of Basin A2 and A6 are due to frontal advance. The areas close to the fronts include partly floating ice so that the observed SEC is smaller than on grounded areas further upstream. The largest loss rates are observed in elevation zones several km inland of the front.

**3.2 Flow velocities, calving fluxes and mass balance by the input/output method**

Data on flow velocities provide on one hand input for deriving calving fluxes, on the other hand information for studying the dynamic response of the glaciers. Figure 3 shows maps of surface velocities in 2011 and 2016, derived from TerraSAR-X and TanDEM-X 11-day repeat pass images, and a map of the difference in velocity between November 1995 and 2016. The 1995 velocity map was derived from interferometric one-day repeat pass data of crossing orbits from the satellites ERS-1 and ERS-2 (map shown in Figure S3 of Rott et al., 2014, Supplementary Material). In

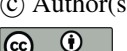


November 1995, ten months after ice shelf collapse, the velocities at calving fronts had already accelerated significantly compared to pre-collapse conditions (Rott et al., 2002). Between 2011 and 2016 the flow velocities slowed down significantly. Even so, in 2016 the terminus velocities of the major outlet glaciers still exceeded the November 1995 velocities.

Details on velocities along central flowlines of Drygalski, Edgeworth and Sjögren glaciers and the position of calving fronts are shown in Figure 4 for different dates between 1995 and 2016. The distance along the x-axis refers to the 1995 grounding line retrieved from ERS-1/ERS-2 InSAR data (Rott et al., 2002). The front of the three glaciers retreated since 1995 by several kilometres, with the largest retreat (11 km) by Sjögren Glacier in 2012. Between 2013 and 2016 the front of Edgeworth Glacier advanced by 1.5 km and the front of Sjögren Glacier by 0.5 km.

Sjögren Glacier shows a gradual decrease of velocity from 2.9 m d$^{-1}$ in August 2009 to 1.5 m d$^{-1}$ in October 2016, referring to the centre of the 2009 front. The velocity on Edgeworth Glacier decreased from 2.5 m d$^{-1}$ in October 2008 to 1.1 m d$^{-1}$ in August 2016. The rate of deceleration between 2013 and 2016 was particularly pronounced on the lowest 6 km of the terminus where the ice was ungrounded. For Drygalski Glacier we show also pre-collapse velocities (January 1993), derived from 35-day ERS-1 repeat pass images by offset tracking. In November 1995 the glacier front was located near the pre-collapse grounding line, but the flow acceleration had already propagated 10 km upstream of the front. Due to rapid flow the phase of the 31 October/1 November 1995 ERS-1/ERS-2 InSAR pair is decorrelated on the lowest two kilometres, prohibiting there interferometric velocity retrieval. Velocities of January 1999 and November 2015 are similar, 7.0 m d$^{-1}$ at the location of the 2015 glacier front. Velocities were lower in 2007 to 2009, and higher in 2011 to 2014, reaching 8.8 m d$^{-1}$ in November 2011.

The recent period of abating flow velocities coincides with years when the sea ice cover persisted during summer. Time series of satellite SAR images show open water in front of the glaciers during several summers up to summer 2008/09 and again in the summers 2010/2011 and 2011/2012. Sea ice persisted all year round from winter 2012 onwards. Open leads in summer and the gradual drift of ice that calved off from the glaciers indicate occasional moderate movement of sea ice.

Slowdown of calving velocities is the main cause for reduced mass deficits during the period 2013 to 2016 compared to previous years. Numbers on calving fluxes for 2011 to 2013 and 2013 to 2016 and the mass balance, derived by the IOM, are specified for four main glacier basins in Table 3. For deriving the calving flux (CF) for each period a linear interpolation between the fluxes at the start date and end date of the period is applied, including a correction for the time lag between ice motion and topography data. If velocity data are available on additional dates in between, these are also taken into account for temporal interpolation. Whereas the SMB between the periods 2011 to



2013 and 2013 to 2016 differs only by 2%, the combined annual calving flux of the four glaciers is
reduced by 16 % during 2013 to 2016 (Table 3). The decrease is even more pronounced when
calving fluxes on individual dates in 2011, 2013 and 2016 are compared. On Drygalski Glacier the
calving flux decreased from 4.03 Gt a$^{-1}$ in November 2011 to 3.34 Gt a$^{-1}$ in December 2013 and
2.92 Gt a$^{-1}$ in September 2016, a decrease by 28 % during the five years.
The differences in the mass balance by TDM SEC (Table 1) and IOM (Table3) are within the
specified uncertainty. For IOM the mass balance of the four glaciers sums up to $B_n$ = -3.26 Gt a$^{-1}$ for
2011 to 2013 and $B_n$ = -2.23 Gt a$^{-1}$ for 2013 to 2016. The corresponding numbers from SEC
analysis, after adding or subtracting the subaqueous mass changes, are $B_n$ = -3.01 Gt a$^{-1}$ and $B_n$ = -
1.99 Gt a$^{-1}$ for the two periods.
For Drygalski Glacier the mass balance numbers for the two periods are $B_n$ = -2.29 Gt a$^{-1}$ and $B_n$ = -
1.80 Gt a$^{-1}$ by IOM, and $B_n$ = -2.18 Gt a$^{-1}$ and $B_n$ = -1.80 Gt a$^{-1}$ (including the subaqueous part) by
TDM SEC analysis. The good agreement of the IOM and SEC mass balance values for Drygalski
Glacier backs up the RACMO estimate for SMB with specific net balance $b_n$ = 1383 kg m$^{-2}$a$^{-1}$. For
the period 1980 to 2016 the mean SMB for Drygalski Glacier by RACMO is 1.35 Gt a$^{-1}$. This is
more than twice the ice mass flux across the grounding line in pre-collapse state (0.58 Gt a$^{-1}$)
obtained as model output by Royston and Gudmundsson (2016) which would imply a highly
positive mass balance taking RACMO SMB as reference for mass input. Velocity measurements in
October/November 1994 at stakes on Larsen A Ice Shelf downstream of Drygalski Glacier show
values that are close to the average velocity of the 10-year period 1984 to 1994 (Rott et al., 1998;
Rack et al., 1999). This supports the assumption that the Larsen A tributary glaciers were
approximately in balanced state before ice shelf collapse.
**4. Elevation change and mass balance of Larsen B glaciers**
**4.1 Elevation change and mass balance by DEM differencing**
The map of surface elevation change dh/dt for the glacier basins discharging into the Larsen B
embayment and SCAR Inlet ice shelf is shown in Figure 5 for the period May/June 2011 to
June/July 2013 and in Figure 6 for June/July 2013 to July/August 2016. The numbers on elevation
change, volume change and mass balance, referring to grounded ice, are specified in Table 4 for
2011 to 2013 and in Table 5 for 2013 to 2016.
The SEC analysis shows large spatial and temporal differences in mass depletion between
individual glaciers. The overall mass deficit of the Larsen B region is dominated by glaciers
draining into the embayment where the ice shelf broke away in 2003 (basins B1 to B11). The annual
mass deficit of the glaciers draining into SCAR Inlet ice shelf (basins B12 to B17) remained modest





and was similar in both periods: $B_n$ = -0.54 Gt a$^{-1}$ during 2011 to 2013 and $B_n$ = -0.58 Gt a$^{-1}$ during
2013 to 2016. The small glaciers (B12 to B15) were in balanced state. The mass balance of Flask
and Leppard glaciers was slightly negative due to flow acceleration and increased ice export after
break-up of the main section of Larsen B Ice Shelf (Wuite et al., 2015).
In 2011 to 2013 the total annual net mass balance of basins B1 to B11 amounted to $B_n$ = -5.75 Gt a$^{-}$
$^{1}$, with the mass deficit dominated by Hektoria-Green (HG) glaciers ($B_n$ = -3.88 Gt a$^{-1}$), followed by
Crane Glacier ($B_n$ = -0.72 Gt a$^{-1}$). The mass losses of Evans and Jorum glaciers and of basin B1
(northeast of Hektoria Glacier) were also substantial, whereas the mass deficit of the other glaciers
was modest. During the period 2013 to 2016 the annual mass deficit of the glacier ensemble was cut
by more than half ($B_n$ = -2.32 Gt a$^{-1}$) compared to 2011 to 2013, with again HG dominating the
mass loss ($B_n$ = -1.54 Gt a$^{-1}$). The decrease in mass depletion was also significant for other glaciers.
For Crane Glacier the 2013 to 2016 loss rate ($B_n$ = -0.22 Gt a$^{-1}$) corresponds to only 18 % of the
estimated balance flux (Rott et al., 2011), a large drop since 2007 with $B_n$ = -3.87 Gt a$^{-1}$ (Wuite et
al., 2015).
The decline of mass depletion coincided with a period of permanent sea ice cover starting in
autumn/winter 2011. During several summers before, including summer 2010/11, the sea ice in
front of the glaciers drifted away and gave way to extended periods with open water. During the
years thereafter the continuous sea ice over obstructed the detachment of frontal ice and facilitated
frontal advance. The maximum terminus advance was observed for HG glaciers, resulting in an
increase of glacier area of 31.6 km$^2$ from 2011 to 2013 and 48.0 km$^2$ from 2013 to 2016 (Table 6).
Due to significant decrease in ice thickness the floating area on Hektoria and Green glaciers
increased significantly after 2011, covering in June 2013 an area of 19.8 km$^2$ and in June 2016 an
area of 62.1 km$^2$ in addition to the frontal advance areas 2011 to 2013, respectively 2013 to 2016,
where the ice was almost completely ungrounded. Areas of floating ice, covering some km$^2$ in area,
were observed on Evans Glacier and Crane Glacier, increasing significantly between 2013 and
2016. The areas of frontal advance showed a similar temporal trend, with an increase of 3.7 km$^2$
between 2011 and 2013 and 5.4 km$^2$ between 2013 and 2016 for Evans Glacier, and 5.0 km$^2$ and
10.5 km$^2$ for Crane Glacier.
Figure 7 shows the altitude dependence of elevation change (dh/dt) for four basins with large mass
deficits. The largest drawdown rate (19.5 m a$^{-1}$) was observed on HG glaciers in the elevation zone
200 m to 300 m a.s.l. during 2011 to 2013, with substantial drawdown up to the 1000 m elevation
zone. On Jorum Glacier the area affected by surface lowering extended up to 700 m elevation, with
a maximum rate of 5 m a$^{-1}$. The drawdown pattern of Crane Glacier is different, with the zone of the
largest 2011 to 2013 drawdown rates (4.5 m a$^{-1}$) commencing about 30 km inland of the front,





extending across the elevation range 500 m to 850 m, abating and shifting further upstream in 2013
to 2016. Scambos et al. (2011) observed an anomalous drawdown pattern on the Crane terminus
during the first few years after ice shelf collapse, very likely associated with drainage of a
subglacial lake.
**4.2 Flow velocities, calving fluxes and mass balance by the input/output method**
Figure 8 shows maps of surface velocities in 2011 and 2016 and a map of the differences in velocity
between November 1995 and 2016. Gaps in the 2011 TerraSAR-X/TanDEM-X velocity map are
filled up with PALSAR data and in the 2016 map with Sentinel-1 data. The 1995 velocity map used
as reference for pre-collapse conditions, was derived from ERS one-day interferometric repeat pass
data. The ERS data show very little difference between 1995 and 1999 flow velocities, suggesting
that the glaciers were close to balanced state during those years (Rott et al, 2011). In 2016 the
velocities of the main glaciers were still higher than in 1995, but had slowed down significantly
since 2011.
The temporal evolution of Larsen B glaciers between 1995 and 2013 is described in detail by Wuite
et al. (2015), showing velocity maps for 1995 and 2008-2012 and time series of velocities along
central flowlines of eight glaciers between 1995 and 2013. In extension, we report here velocity
changes since 2013 and provide details on velocities of HG and Crane glaciers in recent years,
including a diagram of velocities across the flux gates on different dates (Figure 9).
The glaciers discharging into SCAR Inlet ice shelf and the small glaciers of the main Larsen B
embayment (B4, B5, B8 to B11) showed only small variations in velocity since 2011, though in
2016 the velocities of these glaciers were still higher than during the pre-collapse period. The main
glaciers were subject to significant slowdown. On Crane Glacier the velocity in the centre of the
flux gate decreased from a value of 6.8 m d$^{-1}$ in July 2007 to 3.9 m d$^{-1}$ in September 2011, 2.9 m d$^{-1}$
in November 2013 and 2.4 m d$^{-1}$ in October 2016, still 50 % higher than the velocities in 1995 and
1999. Because of major glacier thinning, the cross section of the flux gate decreased significantly,
so that the calving flux amounted in mid-2016 to 1.39 Gt a$^{-1}$, only 20 % larger than in 1995 to 1999.
Since 2007 the drawdown rate of Crane Glacier decreased steadily, from a mass balance $B_n$ = -3.87
Gt a$^{-1}$ in June 2007 to $B_n$ = -0.23 Gt a$^{-1}$ in November 2016. Also on Jorum Glacier the calving
velocity decreased gradually since 2007; during 2013 to 2016 the glacier was close to balanced
state. On the other hand the velocity at the flux gate of Melville Glacier was in 2011 to 2016 only
5 % lower than in 2008, 2.6 times higher than the pre-collapse velocity reported by Rott et al.
(2011). This agrees with the negative mass balance by TDM SEC analysis. However, the mass
deficit is small in absolute terms because of the modest mass turnover.




The velocities of Hektoria and Green glaciers have been subject to significant variations since 2002, associated with major frontal retreat but also intermittent periods of frontal advance (Wuite et al., 2015). Between November 2008 and November 2009 the velocity in the centre of the Hektoria flux gate increased from 1.4 m d$^{-1}$ to 2.8 m d$^{-1}$, slowed down slightly during 2010, and accelerated again in 2011 to reach a value of 4.2 m d$^{-1}$ in November 2011, followed by deceleration to 3.5 m d$^{-1}$ in March 2012, 2.0 m d$^{-1}$ in July 2013 and 1.4 m d$^{-1}$ in June 2016 (Figure 9). Similar deceleration was observed for Green Glacier, from 4.6 m d$^{-1}$ in November 2011, to 2.8 m d$^{-1}$ in July 2013 and 2.0 m d$^{-1}$ in June 2016.

The slowdown and frontal advance of Larsen B calving glaciers coincided with a period of continuous sea ice cover since mid-2011, indicating significant impact of pre-frontal marine conditions on ice flow. Tracking of detached ice blocks close to glacier fronts shows for 2013 to 2016 the following displacements: 6.1 km for Crane Glacier, 2.7 km for Melville Glacier, 2.5 km for Jorum Glacier and 0.9 km for Mapple Glacier. This corresponds to about twice the flux gate velocity for Crane Glacier and about five times for Melville Glacier. The 2013 to 2016 displacement of ice blocks in front of HG glaciers (4.5 km for Green, 3.9 km for Hektoria) exceeded only slightly the distance of frontal advance.

The comparison of mass balance by IOM (Table 7) and SEC shows good overall agreement, as well as for most of the individual basins. The combined 2011 to 2013 annual mass balance of the five basins discharging into the main Larsen B embayment (B2, B3, B6, B7, B10) is $B_n$ = -5.26 Gt a$^{-1}$ by TDM SEC and $B_n$ = -5.63 Gt a$^{-1}$ by IOM, and for 2013 to 2016 $B_n$ = -2.15 Gt a$^{-1}$ by TDM SEC and $B_n$ = -2.28 Gt a$^{-1}$ by IOM. The SEC mass balance in this comparison includes also the volume change of the floating glacier sections (Table 6). Also for Starbuck and Flask glaciers (B13, B16) the mass balance values of the two methods agree well. The only basin where the difference between the two methods exceeds the estimated uncertainty is Leppard Glacier (B17), where IOM ($B_n$ = -0.89 Gt a$^{-1}$ and $B_n$ -0.82 Gt a$^{-1}$ for the two periods) shows larger losses than SEC ($B_n$ = -0.21 Gt a$^{-1}$ and $B_n$ -0.30 Gt a$^{-1}$). The SEC retrievals of the basins B3, B7, B10, B13, B16, which show good agreement between SEC and IOM mass balance, are based on data of the same TDM track as B17. Therefore it can be concluded that the difference in MB of Leppard Glacier is probably due to a bias either in SMB or in the cross section of the flux gate, or in both. The specific surface mass balance (Table 7) for the adjoining Flask Glacier is 39 % higher than for Leppard Glacier.

## 5. Discussion

The main outlet glaciers to the northern sections of Larsen Ice Shelf that disintegrated in 1995 (Prince-Gustav-Channel and Larsen A ice shelves, PGC-LA) and in 2002 (the main section of Larsen B Ice Shelf) are still losing mass due to dynamic thinning. The losses are caused by





accelerated ice flow tracing back to the reduction of backstress after ice shelf break-up triggering
dynamic instabilities (Rott et al., 2002; 2011; Scambos et al., 2004; Wuite et al., 2015; De Rydt et
al., 2015; Royston and Gudmundsson, 2016).
On the outlet glaciers to PGC-LA (basins A1 to A7) the rate of mass depletion of grounded ice
decreased by 40 % from $3.98 \pm 0.33$ Gt a$^{-1}$ during the period 2011 to 2013 to $2.38 \pm 0.18$ Gt a$^{-1}$
during 2013 to 2016. The mass deficit of the area was dominated by losses of Drygalski Glacier,
with annual mass balance $B_n = -2.18$ Gt a$^{-1}$ in 2011 to 2013 and $B_n = -1.72$ Gt a$^{-1}$ in 2013 to 2016.
Scambos et al. (2014) report for 2001 to 2008 a mass change of -5.67 Gt a$^{-1}$ for glacier basins 21 to
25, corresponding approximately to our basins A1 to A7. On Drygalski Glacier the 2003 to 2008
rate of mass depletion ($B_n = -2.39$ Gt a$^{-1}$) by Scambos et al. (2014) was only 9 % higher than our
estimate for 2011 to 2013. On the other glaciers of PGC and Larsen A embayment the slow-down
of calving velocities and decrease in calving fluxes during the last decade was more pronounced.
On the outlet glaciers to Larsen B embayment (basins B1 to B11) the rate of mass depletion for
grounded ice decreased by 60 % from $5.75 \pm 0.45$ Gt a$^{-1}$ during 2011 to 2013 to $2.32 \pm 0.25$ Gt a$^{-1}$
during 2013 to 2016. Hektoria and Green glaciers accounted in both periods for the bulk of the mass
deficit ($B_n = -3.88$ Gt a$^{-1}$, $B_n = -1.54$ Gt a$^{-1}$). High drawdown rates were observed on HG glaciers
during 2011 to 2013, with the maximum value (19.5 m a$^{-1}$) in the elevation zone 200 m to 300 m
a.s.l. Our basins B1 to B11 correspond to the basins 26a and 27 to 31a of Scambos et al. (2014).
Based on ICESat data spanning September 2003 to March 2008 and optical stereo image DEMs
acquired between November 2001 to November 2006, Scambos et al. (2014) report for these basins
an annual mass balance $B_n = -8.39$ Gt a$^{-1}$ excluding ice lost by frontal retreat. Our rate of mass loss
for 2011 to 2013 amounts to 69% of this value, and for 2013 to 2016 to 36%, a similar percentage
decrease of mass losses as for the PGC-LA basins. After ice shelf break-up in March 2002 glacier
flow accelerated rapidly, causing large increase of calving fluxes during the first years after Larsen
B collapse, whereas on most glaciers the calving velocities slowed down significantly after 2007
(Scambos et al., 2004, 2011; Rott et al., 2011; Shuman et al., 2011; Wuite at al., 2015). An
exception is basin B2 (HG glaciers) for which the 2011 to 2013 loss rate was 2% higher than the
value ($B_n = -3.82$ Gt a$^{-1}$) reported by Scambos et al. (2014) for 2001 to 2008.
The drawdown pattern on the main glaciers shows high elevation loss rates for grounded ice shortly
upstream of the glacier front or upstream of the floating glacier section, and abating loss rates
towards higher elevation. This is the typical loss pattern for changes in the stress state at the
downstream end of a glacier as response to the loss of terminal floating ice (Hulbe et al., 2008). The
elevation change pattern of recent years is different on Crane Glacier, where elevation decline and
thinning migrated up-glacier during 2011 to 2016, an indication for upstream-propagating





disturbances (Pfeffer, 2007). Both patterns indicate that the glaciers are still away from equilibrium
state and dynamic thinning will continue for years.
We compiled surface motion and calving fluxes for main glaciers of the study region and derived
the surface mass balance from output of the regional atmospheric climate model RACMO. These
data enable to compare individual components of the mass balance. Whereas the SMB differed
between the periods 2011 to 2013 and 2013 to 2016 only by few per cent, the calving fluxes
decreased significantly due to slow-down of ice motion, confirming that the mass losses were of
dynamic origin, an aftermath to changes in the stress regime after ice shelf collapse.
The terminus velocities on most glaciers are still higher than during the pre-collapse period. After
rapid flow acceleration during the first years after ice shelf break-up there has been a general trend
of deceleration afterwards, however with distinct differences in the temporal pattern between
individual glaciers. Glaciers with broad calving fronts show larger temporal variability of velocities
and calving fluxes than glaciers with small width to length ratio. In the Larsen A embayment the
Drygalski Glacier has been subject to major variations in flow velocity and calving flux during the
last decade. In 2007 to 2009 the velocity in the centre of the flux gate varied between 5.5 m d$^{-1}$ and
6 m d$^{-1}$, increased to 8 m d$^{-1}$ in 2011 and 2012, and decreased to 6.0 m d$^{-1}$ in July 2016, still four
times higher than the velocity in 1993. In the Larsen B embayment Hektoria and Green glaciers
showed large temporal fluctuation in velocity and a general trend of frontal retreat, but also sporadic
periods of frontal advance. A major intermittent acceleration event, starting in 2010, was
responsible for a large mass deficit in 2011 to 2013.
Regarding the SCAR Inlet ice shelf tributaries, the small glaciers (basin B12 to B15) were
approximately in balanced state, whereas Flask (B16) and Leppard (B17) glaciers had a moderate
mass deficit. The total mass balance of the SCAR Inlet glaciers, based on TDM SEC analysis, was
$B_n$ = -0.54 ± 0.38 Gt a$^{-1}$ in 2011 to 2013 and $B_n$ = -0.58 ± 0.38 Gt a$^{-1}$ in 2013 to 2016. As for the
calving glaciers to the Larsen A and B embayments, the loss rate was lower than during the period
2001 to 2008 ($B_n$ = -1.37 Gt a$^{-1}$) reported by Scambos et al. (2014).
The slowdown of flow velocities and decline in mass depletion between 2011 and 2016 coincided
with periods of continuous sea ice cover. After several summers with open water (excluding
summer 2009/10 when sea ice persisted), a period of permanent sea ice cover in front of the glaciers
commenced in Larsen B embayment in winter 2011 and in PGC and Larsen A embayment in winter
2013. The sea ice cover impeded glacier calving, as apparent in frontal advance of several glaciers.
Large frontal advance was observed for HG glaciers (~3.2 km during 2011 to 2013 and ~3.8 km
during 2013 to 2016) and Crane Glacier (~1.2 km during 2011 to 2013 and ~2.5 km during 2013 to



2016). The front of Bombardier-Edgeworth glaciers advanced between 2013 and 2016 by 1.5 km
and the front of Sjögren Glacier by 0.5 km. The continuous sea ice cover and restricted movement
of ice calving off from glaciers contrasts with the rapid movement of icebergs during the first few
days after Larsen A and B collapse, drifting away by up to 20 km per day due to strong downslope
winds and local ocean currents (Rott et al., 1996; Rack and Rott 2004). For 2006 to 2015 a modest
trend of atmospheric cooling was observed in the study region, in particular in summer (Turner et
al., 2016; Oliva et al., 2017). However, this feature does not fully explain the striking difference in
sea ice pattern and ice drift.
**6. Conclusions**
The analysis of surface elevation change by DEM differencing over the periods 2011 to 2013 and
2013 to 2016 shows continuing drawdown and major losses in ice mass for outlet glaciers to Prince-
Gustav-Channel and the Larsen A and B embayments. During the observation period 2011 to 2016
there was a general trend of decreasing mass depletion, induced by slowdown of calving velocities
resulting in reduced calving fluxes. For several glaciers frontal advance was observed in spite of
ongoing elevation losses upstream. The mass balance numbers for the glaciers north of Seal
Nunataks are $B_n = -3.98 \pm 0.33$ Gt a$^{-1}$ during 2011 to 2013 and $B_n = -2.38 \pm 0.18$ Gt a$^{-1}$ during 2013
to 2016. The corresponding numbers for glaciers calving into the Larsen B embayment for the two
periods are $B_n = -5.75 \pm 0.45$ Gt a$^{-1}$ and $B_n = -2.32 \pm 0.25$ Gt a$^{-1}$. For the glacier discharging into
SCAR Inlet ice shelf the losses were modest.
The period of decreasing flow velocities and frontal advance coincides with several years when the
sea ice cover persisted during summer. Considering the ongoing mass depletion and the increase of
ungrounded glacier area due to thinning, we expect recurrence of periods with frontal retreat and
increasing calving fluxes, in particular for those glaciers that showed major temporal variations in
ice flow during the last several years.
In Larsen A embayment large fluctuations in velocity were observed for Drygalski Glacier, and in
Larsen B embayment for Hektoria and Green glaciers. These are the glaciers with the main share in
the overall mass losses of the region: Drygalski Glacier contributed 61 % to the 2011 to 2016 mass
deficit of the Larsen A/PGC outlet glaciers, and HG glaciers accounted for 67 % of the mass deficit
of the Larsen B glaciers. On HG glaciers the ice flow accelerated significantly in 2010/2011,
triggering elevation losses up to 19.5 m a$^{-1}$ on the lower terminus during the period 2011 to 2013.
HG glaciers have a joint broad calving front and the frontal sections are ungrounded, thus being
more vulnerable to changes in atmospheric and oceanic boundary conditions than glaciers that are
confined in narrow valleys.



Complementary to DEM differencing, we applied the input/output method to derive the mass
balance of the main glaciers. The mass balance numbers of these two independent methods show
good agreement, affirming the soundness of the reported results. The agreement backs up also the
reliability of the RACMO SMB data. A strong indicator for the good quality of the TDM SEC
products is the good agreement with 2011-2016 SEC data measured by the airborne laser scanner of
NASA IceBridge. Both data sets were independently processed. The agreement indicates that SAR
signal penetration does not affect the retrieval of surface elevation change on glaciers by InSAR
DEM differencing if repeat observation data are acquired over snow/ice media with stable
backscatter properties under the same observation geometry.

*Data availability*. Data sets used in this study will be made available upon publication of the final
version on cryoportal.enveo.at.
*Competing interests*. The authors declare that they have no conflict of interest.

*Acknowledgements*. The TerraSAR-X data and TanDEM-X data were made available by DLR
through projects HYD1864, XTI_GLAC1864, XTI_GLAC6809 and DEM_GLA1059. Sentinel- 1
data were obtained through the ESA Sentinel Scientific Data Hub, ALOS PALSAR data through the
ESA ALDEN AOALO 3741 project. Landsat 8 images, available at USGS Earth Explorer, were
downloaded via Libra browser. The IceBridge ATM L4 Surface Elevation Rate of Change and
IceBridge MCoRDS Ice Thickness version V001 data were downloaded from the NASA
Distributed Active Archive Center, US National Snow and Ice Data Center (NSIDC), Boulder,
Colorado. We wish to thank A. Cook (Univ. Swansea, UK) for providing outlines of glacier basins.
The work was supported by the European Space Agency, ESA Contract No. 4000115896/15/I-LG,
High Resolution SAR Algorithms for Mass Balance and Dynamics of Calving Glaciers (SAMBA).

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




## Tables

**Table 1**. Rates of surface elevation change, volume change and mass balance by means of TDM DEM differencing 2013 to 2016, for glacier basins discharging into Prince-Gustav-Channel, Larsen Inlet and Larsen A embayment. dh/dt is the mean rate of elevation change of the area covered by the high resolution map (Fig. 1). The basin area refers to ice front positions delineated in TanDEM-X images of 2016-07-16, 2016-07-27, 2016-08-18. The rates of ice volume change (dV/dt) and total mass balance (dM/dt) refer to grounded ice. *dM/dt 2011-2013 for grounded areas of basins A1 to A7 from the TDM SEC analysis by Rott et al., (2014).

| ID | Basin name | Basin area [km$^2$] | dh/dt map [km$^2$] | dh/dt [m a$^{-1}$] | dV/dt [km$^3$ a$^{-1}$] | Uncertainty [km$^3$ a$^{-1}$] | dM/dt [Gt a$^{-1}$] 2013-16 | *dM/dt [Gt a$^{-1}$] 2011-13 |
|---|---|---|---|---|---|---|---|---|
| A1 | Cape Longing Peninsula | 668.9 | 576.9 | -0.257 | -0.146 | ±0.041 | -0.131 | -0.150 |
| A2 | Sjögren-Boydell (SB) | 527.6 | 188.0 | -1.239 | -0.241 | ±0.046 | -0.217 | -0.364 |
| A3 | APPE glaciers | 513.6 | 231.9 | -0.137 | -0.032 | ±0.052 | -0.029 | +0.056 |
| A4 | DBE glaciers | 653.9 | 194.3 | -0.286 | -0.063 | ±0.058 | -0.057 | -0.396 |
| A5 | Sobral Peninsula | 257.9 | 198.5 | -0.173 | -0.034 | ±0.018 | -0.031 | -0.145 |
| A6 | Cape Worsley coast | 625.1 | 291.4 | -0.742 | -0.217 | ±0.051 | -0.195 | -0.800 |
| A7 | Drygalski Glacier | 998.3 | 604.7 | -3.187 | -1.913 | ±0.074 | -1.722 | -2.179 |
| | *Total* | *4245.3* | *2285.7* | | *-2.646* | *±0.199* | *-2.382* | *-3.978* |

**Table 2.** (a) Area extent of floating ice in 2016; (b) and (c) rate of surface elevation change and volume change 2013 to 2016 of floating ice (excluding the areas of frontal advance); (d) and (e) extent and volume of frontal advance (+) or retreat (-) areas.

| ID | Basin name | (a) Floating area [km$^2$] | (b) Mean dh/dt [m a$^{-1}$] | (c) Mean dV/dt [km$^3$ a$^{-1}$] | (d) Advance/ retreat area [km$^2$] | (e) Volume [km$^3$] |
|---|---|---|---|---|---|---|
| A2 | Sjögren-Boydell | 6.09 | +1.250 | 0.062 | +1.96 | +0.403 |
| A4 | DBE glaciers | 56.22 | +0.131 | 0.060 | +11.74 | +2.017 |
| A6 | Cape Worsley coast | 4.89 | +0.194 | 0.008 | +2.92 | +0.550 |
| A7 | Drygalski Glacier | 4.57 | -2.231 | -0.082 | -1.40 | -0.360 |





**Table 3.** Mean specific surface mass balance, $b_n$, for 2011 to 2016, and rates of surface mass
balance (SMB), calving flux (CF) and mass balance by IOM (MB) in Gt a$^{-1}$ for the periods 2011 to
2013 and 2013 to 2016 for outlet glaciers north of Seal Nunataks.

| ID | Glacier | $b_n$ 11-16 kg m$^{-2}$a$^{-1}$ | SMB 2011-13 Gt a$^{-1}$ | SMB 2013-16 Gt a$^{-1}$ | CF 2011-13 Gt a$^{-1}$ | CF 2013-16 Gt a$^{-1}$ | MB 2011-13 Gt a$^{-1}$ | MB 2013-16 Gt a$^{-1}$ |
|---|---|---|---|---|---|---|---|---|
| A2 | SB | 653 | 0.314 | 0.362 | 0.861 | 0.673 | -0.547±0.144 | -0.311±0.119 |
| A3 | APPE | 903 | 0.446 | 0.470 | 0.517 | 0.488 | -0.071±0.088 | -0.018±0.089 |
| A4 | DBE | 982 | 0.624 | 0.646 | 0.980 | 0.748 | -0.356±0.181 | -0.102±0.153 |
| A7 | Drygalski | 1383 | 1.398 | 1.374 | 3.687 | 3.177 | -2.289±0.619 | -1.803±0.544 |

**Table 4**. Rate of surface elevation change for areas by means of TDM DEM differencing 2011 to
2013 for glacier basins of the Larsen B embayment. dh/dt is the mean rate of elevation change of
the area covered by the high resolution map (Fig. 5). The basin area refers to ice front positions
delineated in TanDEM-X images of 2013-06-20 and 2013-07-01. The rates of ice volume change
(dV/dt) and total mass balance (dM/dt) refer to grounded ice.

| ID | Basin name | Total basin area [km²] | TDM surveyed area [km²] | Mean dh/dt [m a$^{-1}$] | dV/dt [km³ a$^{-1}$] | Uncertainty [km³ a$^{-1}$] | dM/dt [Gt a$^{-1}$] |
|---|---|---|---|---|---|---|---|
| B1 | West of SN | 638.1 | 494.1 | -0.693 | -0.342 | ±0.063 | -0.308 |
| B2 | Hektoria Green | 1167.5 | 491.8 | -8.844 | -4.312 | ±0.145 | -3.881 |
| B3 | Evans | 266.9 | 137.3 | -2.700 | -0.364 | ±0.032 | -0.328 |
| B4 | Evans Headland | 117.7 | 106.8 | -0.476 | -0.051 | ±0.011 | -0.046 |
| B5 | Punchbowl | 119.9 | 84.2 | -0.761 | -0.064 | ±0.013 | -0.058 |
| B6 | Jorum | 460.3 | 110.6 | -2.157 | -0.239 | ±0.063 | -0.215 |
| B7 | Crane | 1322.6 | 343.8 | -2.318 | -0.805 | ±0.179 | -0.724 |
| B8 | Larsen B coast | 142.6 | 95.8 | -0.085 | -0.046 | ±0.016 | -0.041 |
| B9 | Mapple | 155.4 | 92.4 | -0.524 | -0.048 | ±0.018 | -0.043 |
| B10 | Melville | 291.5 | 139.9 | -0.859 | -0.120 | ±0.036 | -0.108 |
| B11 | Pequod | 150.3 | 115.1 | +0.025 | +0.003 | ±0.015 | +0.003 |
| | *Total B1 to B11* | *4832.9* | *2211.6* | | *-6.388* | *±0.495* | *-5.749* |
| B12 | Rachel | 51.8 | 38.9 | -0.046 | -0.002 | ±0.006 | -0.002 |
| B13 | Starbuck | 299.4 | 169.4 | -0.118 | -0.020 | ±0.035 | -0.018 |
| B14 | Stubb | 108.3 | 87.9 | +0.116 | -0.001 | ±0.011 | -0.001 |
| B15 | SCAR IS coast | 136.8 | 102.4 | -0.184 | -0.019 | ±0.014 | -0.017 |
| B16 | Flask | 1130.6 | 516.3 | -0.629 | -0.325 | ±0.138 | -0.292 |
| B17 | Leppard | 1851.0 | 946.5 | -0.243 | -0.230 | ±0.219 | -0.207 |
| | *Total B12 to B17* | *3577.9* | *1861.4* | | *-0.597* | *±0.423* | *-0.537* |



**Table 5**. Rate of surface elevation change for areas by means of TDM DEM differencing 2013 to
2016 for glacier basins of the Larsen B embayment. dh/dt is the mean rate of elevation change of
the area covered by the high resolution map (Fig. 6). The basin area refers to ice front positions
delineated in TanDEM-X images of 2016-06-27 and 2016-08-01. The rates of ice volume change
(dV/dt) and total mass balance (dM/dt) refer to grounded ice.

| ID | Basin name | Total basin area [km²] | TDM surveyed area [km²] | Mean dh/dt [m a⁻¹] | dV/dt [km³ a⁻¹] | Uncertainty [km³ a⁻¹] | dM/dt [Gt a⁻¹] |
|---|---|---|---|---|---|---|---|
| B1 | West of SN | 638.7 | 485.6 | -0.172 | -0.084 | ±0.043 | -0.076 |
| B2 | Hektoria Green | 1215.7 | 552.8 | -3.092 | -1.708 | ±0.099 | -1.538 |
| B3 | Evans | 272.3 | 165.3 | -1.494 | -0.238 | ±0.021 | -0.214 |
| B4 | Evans Headland | 117.7 | 106.8 | -0.331 | -0.035 | ±0.007 | -0.032 |
| B5 | Punchbowl | 119.9 | 84.2 | -0.488 | -0.041 | ±0.009 | -0.037 |
| B6 | Jorum | 461.4 | 111.7 | -0.989 | -0.110 | ±0.042 | -0.099 |
| B7 | Crane | 1333.4 | 354.0 | -0.753 | -0.247 | ±0.120 | -0.222 |
| B8 | Larsen B coast | 142.6 | 96.0 | -0.166 | -0.016 | ±0.011 | -0.014 |
| B9 | Mapple | 155.4 | 92.8 | -0.240 | -0.022 | ±0.012 | -0.020 |
| B10 | Melville | 292.9 | 140.9 | -0.584 | -0.081 | ±0.024 | -0.073 |
| B11 | Pequod | 150.6 | 115.3 | +0.069 | 0.008 | ±0.011 | +0.007 |
| | *Total B1 to B11* | *4900.2* | *2305.5* | | *-2.574* | *±0.335* | *-2.318* |
| B12 | Rachel | 51.8 | 38.9 | +0.040 | 0.002 | ±0.004 | +0.002 |
| B13 | Starbuck | 299.4 | 169.4 | +0.006 | 0.001 | ±0.023 | +0.001 |
| B14 | Stubb | 108.3 | 87.9 | +0.115 | 0.010 | ±0.007 | +0.009 |
| B15 | SCAR IS coast | 136.8 | 102.4 | -0.087 | -0.009 | ±0.009 | -0.008 |
| B16 | Flask | 1130.6 | 516.3 | -0.604 | -0.312 | ±0.092 | -0.281 |
| B17 | Leppard | 1851.0 | 946.5 | -0.345 | -0.337 | ±0.146 | -0.303 |
| | *Total B12 to B17* | *3577.9* | *1861.5* | | *-0.645* | *±0.281* | *-0.580* |








**Table 6.** (a) Area extent of floating ice in 2013 (A) and 2016 (B); (b) and (c) rate of surface
elevation change and volume change 2011 to 2013 (A) and 2013 to 2016 (B) of floating ice
(excluding the areas of frontal advance); (d) and (e) extent and volume of frontal advance areas.

| ID | Basin name | (a) Floating area [km$^2$] | (b) Mean dh/dt [m a$^{-1}$] | ( c) Mean dV/dt [km$^3$ a$^{-1}$] | (d) Advance area [km$^2$] | (e) Volume [km$^3$] |
|---|---|---|---|---|---|---|
| **(A) 2011 - 2013** | | | | | | |
| B2 | HG | 19.81 | -1.920 | -0.308 | 31.65 | 11.676 |
| B3 | Evans | 5.55 | -1.264 | -0.057 | 3.66 | 0.807 |
| B6 | Jorum | 0.40 | +3.510 | +0.011 | 0.54 | 0.134 |
| B7 | Crane | 2.01 | +3.770 | +0.061 | 4.96 | 2.164 |
| **(B) 2013 - 2016** | | | | | | |
| B2 | HG | 62.09 | -0.002 | -0.001 | 47.96 | 11.270 |
| B3 | Evans | 14.56 | -0.652 | -0.077 | 5.39 | 0.931 |
| B6 | Jorum | 1.15 | +0.305 | +0.003 | 0.78 | 0.165 |
| B7 | Crane | 7.99 | -2.620 | -0.169 | 10.54 | 3.301 |
| B10 | Melville | 0.88 | -0.966 | -0.007 | 1.20 | 0.219 |


**Table 7.** Mean specific surface mass balance (b$_n$) 2011-2016, annual surface mass balance (SMB)
and calving flux (CF) 2011-2013 and 2013-2016, and resulting IOM mass balance (MB) in Gt a$^{-1}$
for Larsen B glaciers.

| ID | Glacier | b$_n$ 11-16 kg m$^{-2}$a$^{-1}$ | SMB 2011-13 Gt a$^{-1}$ | SMB 2013-16 Gt a$^{-1}$ | CF 2011-13 Gt a$^{-1}$ | CF 2013-16 Gt a$^{-1}$ | MB 2011 -13 Gt a$^{-1}$ | MB 2013 -16 Gt a$^{-1}$ |
|---|---|---|---|---|---|---|---|---|
| B2 | HG | 1400 | 1.563 | 1.644 | 5.733 | 3.389 | -4.170±0.936 | -1.745±0.590 |
| B3 | Evans | 562 | 0.137 | 0.156 | 0.389 | 0.304 | -0.252±0.065 | -0.148±0.053 |
| B6 | Jorum | 884 | 0.376 | 0.427 | 0.457 | 0.361 | -0.081±0.092 | +0.066±0.86 |
| B7 | Crane | 837 | 1.023 | 1.159 | 2.093 | 1.565 | -1.070±0.280 | -0.406±0.247 |
| B10 | Melville | 330 | 0.091 | 0.100 | 0.146 | 0.144 | -0.055±0.021 | -0.044±0.022 |
| B13 | Starbuck | 287 | 0.078 | 0.091 | 0.067 | 0.068 | +0.011±0.014 | +0.023±0.016 |
| B16 | Flask | 693 | 0.722 | 0.824 | 1.085 | 1.118 | -0.363±0.163 | -0.294±0.176 |
| B17 | Leppard | 500 | 0.874 | 0.961 | 1.760 | 1.780 | -0.886±0.237 | -0.819±0.246 |







**Figures**

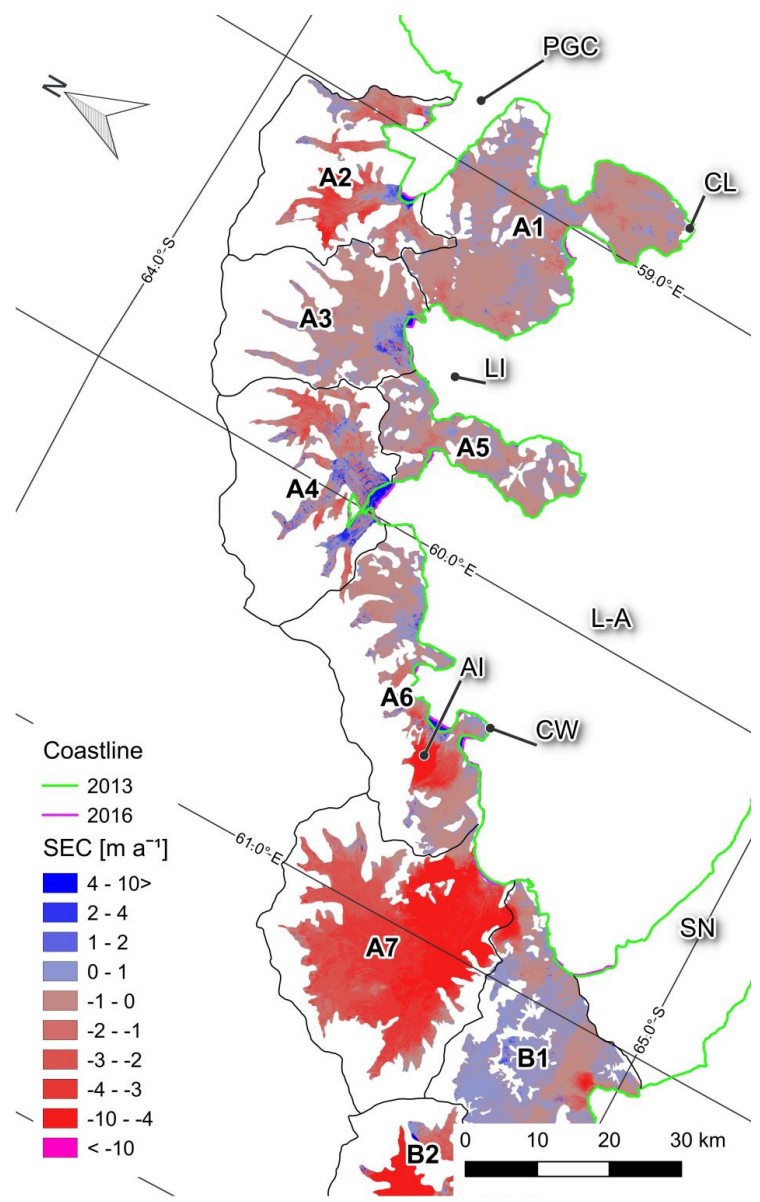


**Figure 1**. Map of surface elevation change dh/dt (m a$^{-1}$) June/July 2013 to July/August 2016 on
glaciers north of Seal Nunataks (SN). AI – Arrol Icefall, CL – Cape Longing, CW – Cape Worsley,.
L-A – Larsen A embayment, LI – Larsen Inlet, PGC – Prince-Gustav-Channel.



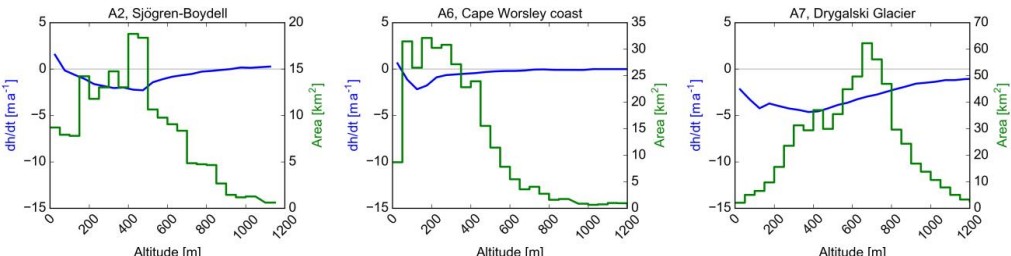


**Figure 2.** Rate of glacier surface elevation change dh/dt (in m a$^{-1}$) 2013 to 2016 versus altitude in

50 m intervals for basins A2, A6 and A7. Green line: hypsometry of surveyed glacier area in km$^2$.

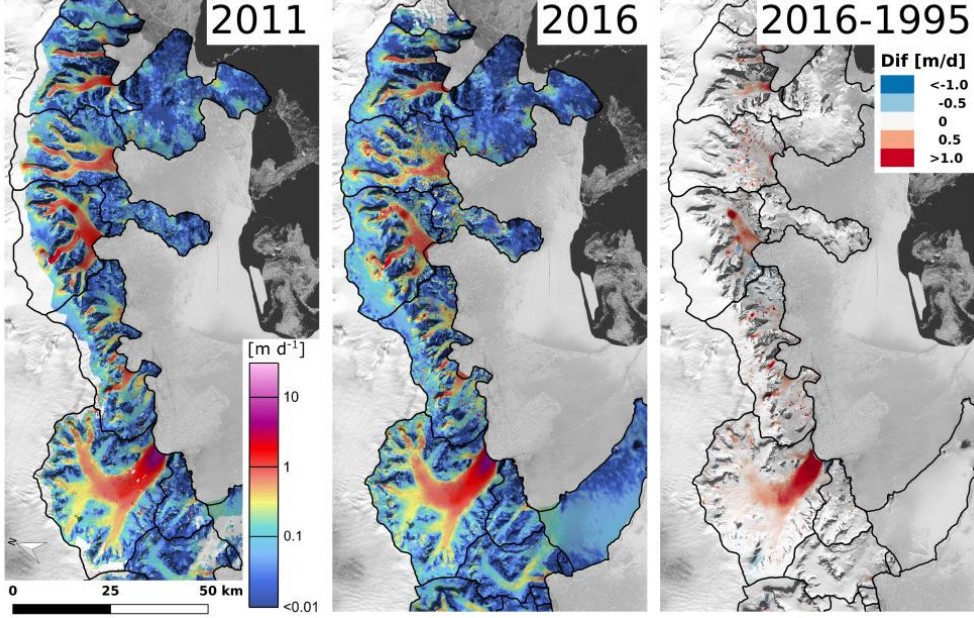


**Figure 3.** Magnitude of ice velocity [m d$^{-1}$] 2011 and 2016 derived from TerraSAR-X and

TanDEM-X data. Gaps in 2011 filled with PALSAR data and in 2016 filled with Sentinel-1 data.
Right: Map of velocity difference 2016 minus 1995 (November).





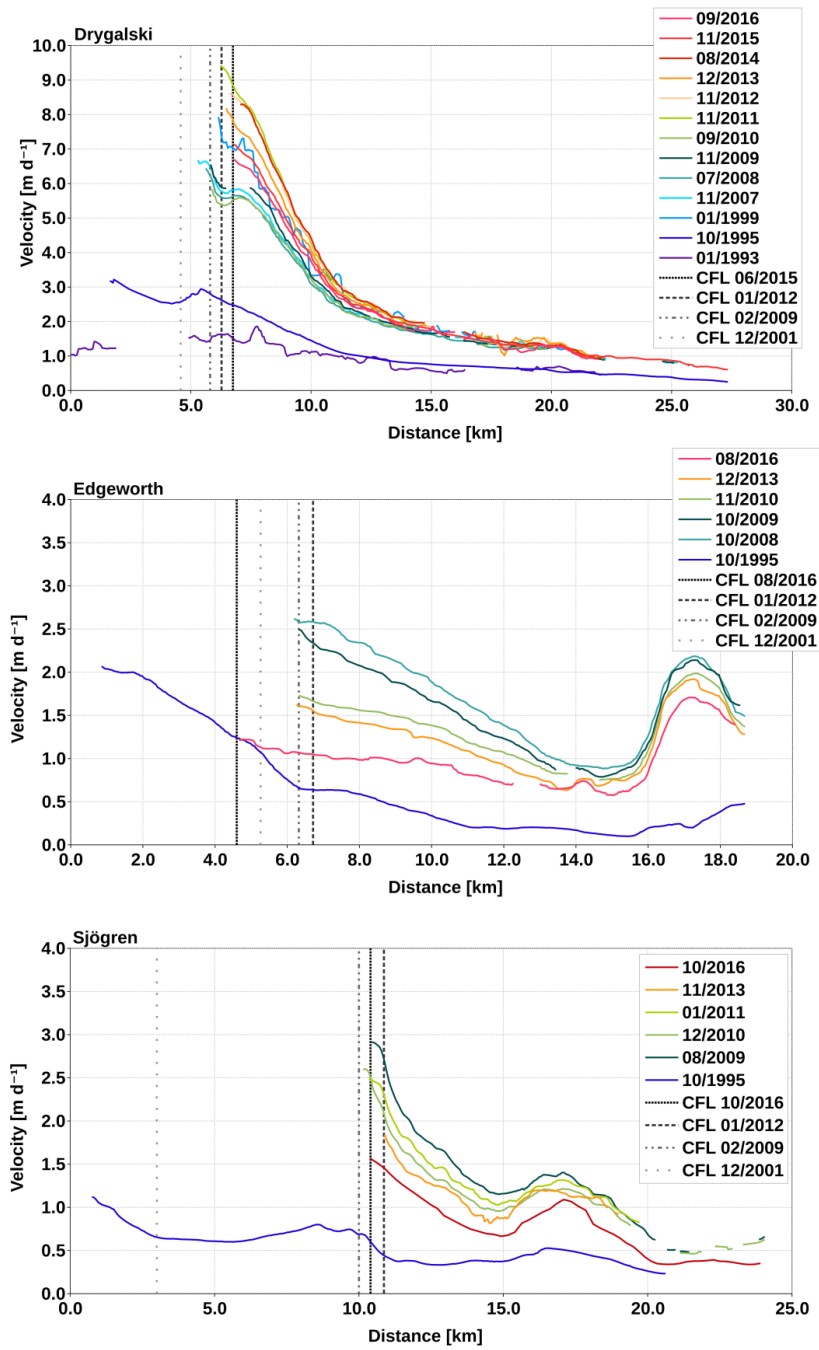


**Figure 4**. Surface velocities along the central flow lines of Drygalski, Edgeworth and Sjögren
glaciers and their frontal positions on different dates (month/year). The x- and y-scales are different
for individual glaciers. Vertical lines show positions of the calving front.




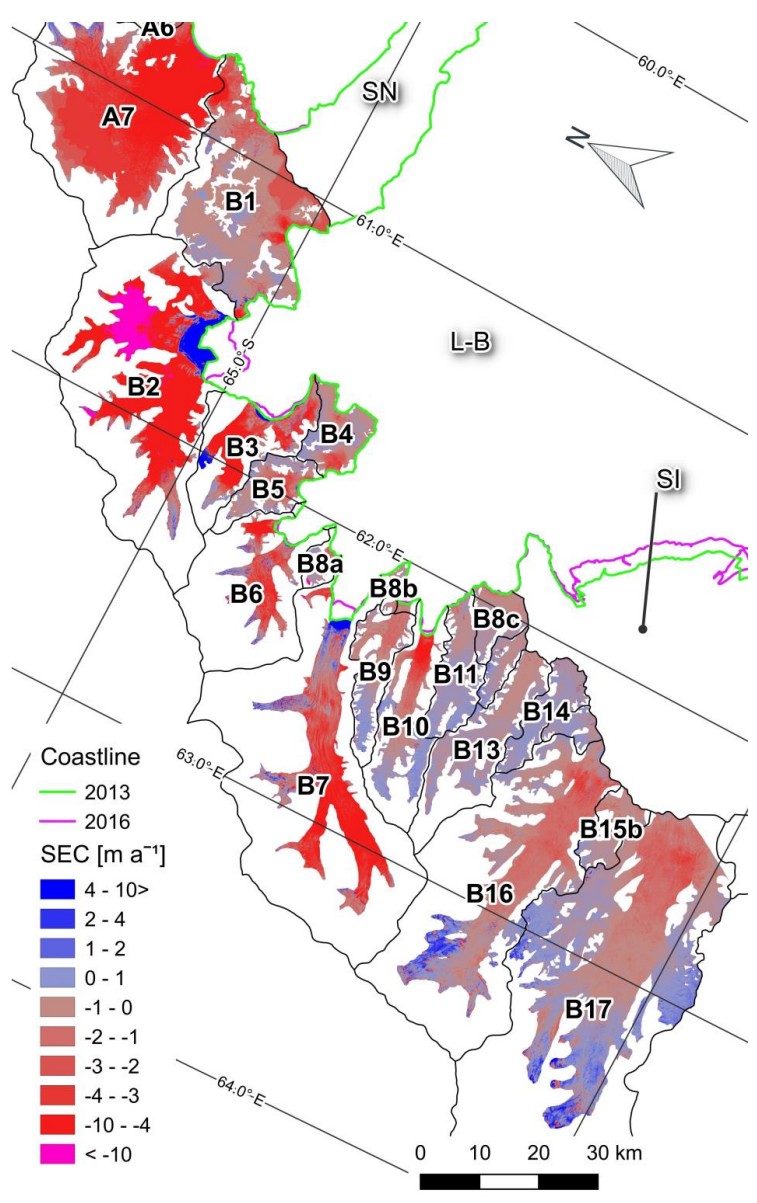


**Figure 5**. Map of surface elevation change (SEC m a$^{-1}$) May/June 2011 to June/July 2013 on glaciers of Larsen B embayment (L-B). SN – Seal Nunataks. SI -SCAR Inlet ice shelf.





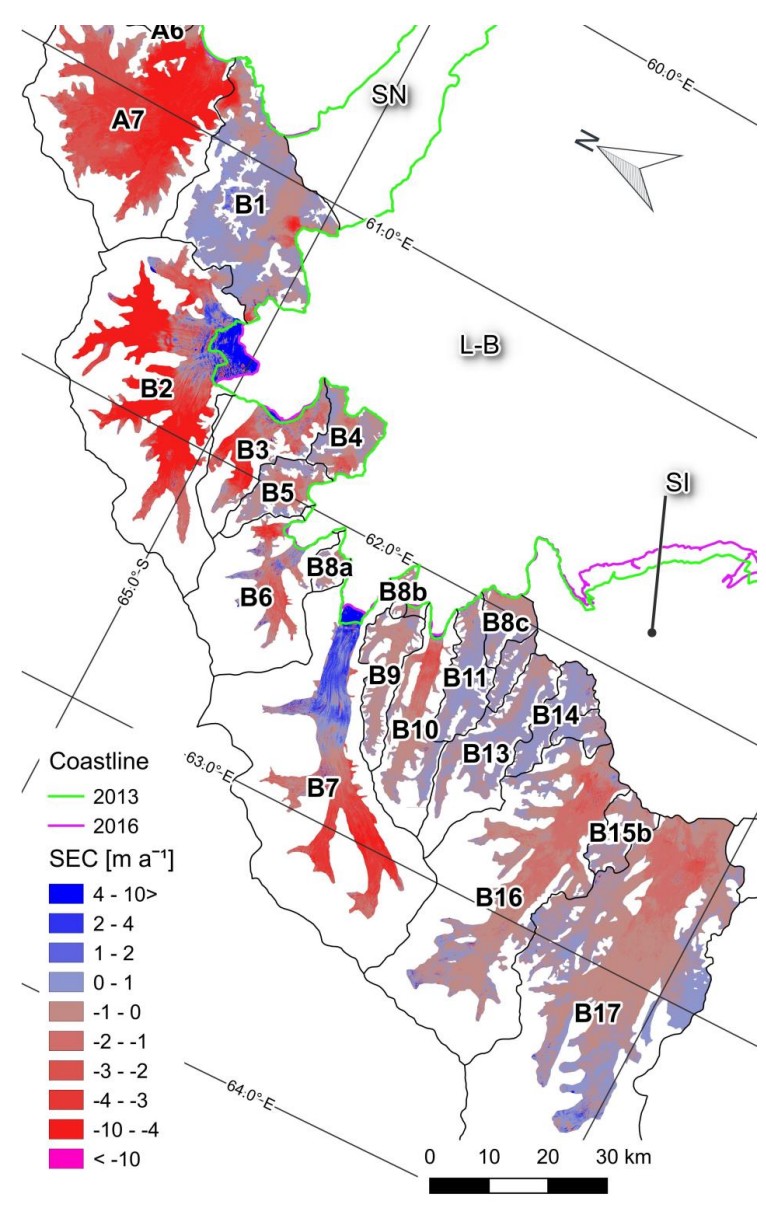


**Figure 6**. Map of surface elevation change (SEC m a⁻¹) June/July 2013 to July/August 2016 on
glaciers of Larsen B embayment (L-B). SN – Seal Nunataks. SI -SCAR Inlet ice shelf.









**Figure 7.** Rate of glacier surface elevation change dh/dt (in m a$^{-1}$) 2011 to 2013 and 2103 to 2016

versus altitude in 50 m intervals for basins B2. B6. B7 and B10. Green line: hypsometry of

surveyed glacier area in km$^2$.





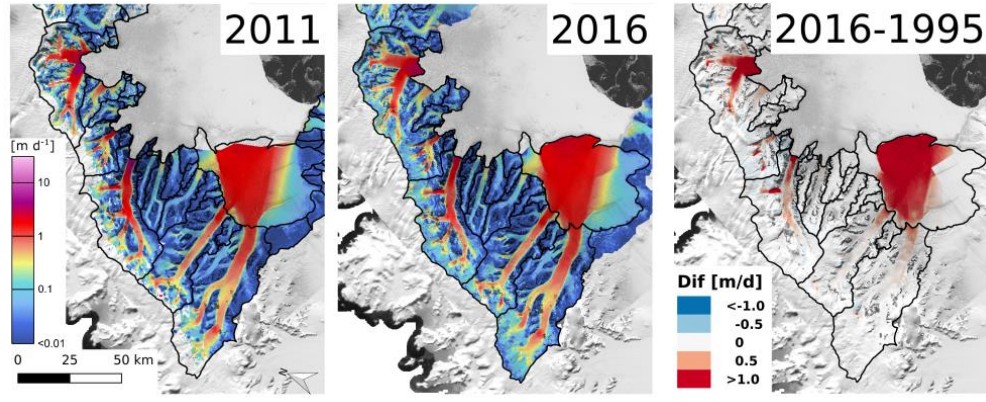

**Figure 8.** Magnitude of ice velocity [m d$^{-1}$] 2011 and 2016 derived from TerraSAR-X and TanDEM-X data. Gaps in 2011 filled with PALSAR data and in 2016 filled with Sentinel-1 data. Right: Map of velocity difference 2016 minus 1995.

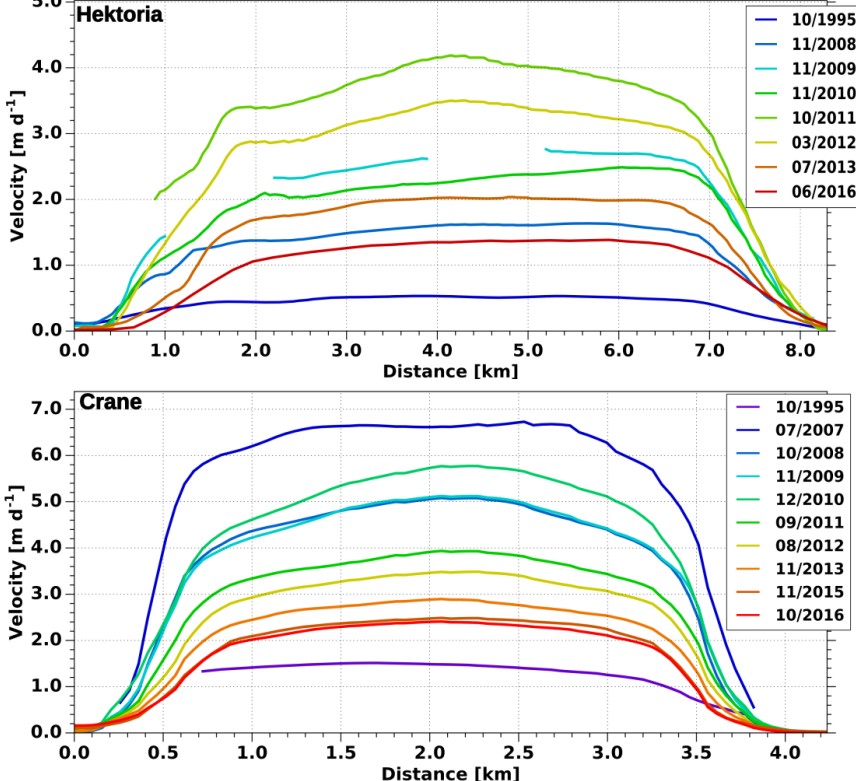

**Figure 9.** Surface velocity across the flux gate of Hektoria Glacier and Crane Glacier on different dates (month/year) between 1995 and 2016.