# Peer review of "Changing pattern of ice flow and mass balance for glaciers discharging into the Larsen A and"

_The Cryosphere, 2017_

## Referee Comment (RC1) · Anonymous Referee #1 · 27 Dec 2017

Review of Rott et al., TCryoD : Changing pattern of ice flow and mass balance. . . Larsen A and B. . .

The paper presents the results of a new analysis of elevation change and flow speed change for the eastern Antarctic Peninsula from Sjogren-Boydell glacier to Leppard Glacier, spanning the major outlets on the mainland Peninsula that were affected by the loss of ice shelves in 1995 and 2002. The study shows that the systems have moved in a positive direct in mass balance (either less negative, or positive outright) in the past few years. They attribute the decline of loss rate to the persistent presence of fast ice in the embayments.

[Figure]

This is a very clear and well-written study, with a lot of good (accurate) new data to offer. It could be published as it is. It provides a 'next chapter' in the monitoring of this rapidly-changing region impacted by ∼25 years of very warm conditions (1980-2006) which have tapered to slightly cooling over the past several years (still warmer than the mid-20th century by a considerable amount). Even at this point, two decades past the ice shelf disintegrations, the glacier systems still show short-term changes in both elevation and flow speed.

Attribution of the reduction in ice losses (less negative mass balance, and in some cases a switch to positive mass balance) is given to a reduction in calving flux – i.e. a downstream movement of the calving front. This in turn is attributed to persistent fast ice.

MInor comments follow. I would encourage the authors to adopt a consistent sign convention for mass budget / mass balance, i.e. negative means ice mass leaving the system. Although there is not a great deal of ambiguity (the words and numbers match the meaning everywhere I have checked), in places one wonders if a positive 'loss' might mean a net gain or a net loss, etc.

L22 please use 'mass budget method' rather than 'input-output' method. Note that IMBIE-2 has now adopted this phrasing. L32 no need to preface the mass budget results with 'Bn =' – it's a bit confusing, since you have not introduced that variable name, and not necessary. Also – L34 a 'mass loss' for a glacier losing mass would be –positive- : these are mass balance results, so negative numbers already mean 'loss'. A picky point, but this has been made strenuously by other authors/speakers. L41 this 'sea ice cover' was/is 'landfast ice' – another picky point perhaps, but an upcoming paper will discuss this buttressing, and fast ice is a much better buttress than typical sea ice. L68 here 'loss' is positive, as it should be with the phrasing; but better to stick to one numerical convention, positive or negative, and use words accordingly. It looks as though the majority of the Introduction uses positive numbers to report 'mass loss', and that is appropriate. However, it might be a bit confusing to people, since in
terms of 'mass balance' these numbers should all be negative. In any case, please be consistent throughout the paper (abstract differs from main text). See L89-L100 and elsewhere. L76 change to '. . .began to accelerate and thin. . .' ('get thinner' is a bit colloquial, almost slang) L154 'data takes' is also a bit colloquial; 'swath data' or 'data acquisitions'? L162, L165, L166 I think that "Raw DEMs" should be "raw DEMs" in English convention. L173 change to '. . .data from the Antarctic Peninsula DEM. . ..' L181 'swaths', not swathes (in US English at least) L219 remove 'anywhere', and change to 'back-slope areas' - this is slightly confusing on first read. L254-255 – the RMSD is somewhat high, 50 – 60 m/yr, though, a bit of a concern. L260 again, please change to 'mass budget method'. L297 '. . .approximately in a balanced state. . .' ; Same Note for L365. L311-312 see comment re Figures 3 and 8. L510-513 – it would seem that several of the longer, thinner glaciers are evolving toward the Crane pattern of elevation change – the DBE system as well as Sjogren Glacier; and in the latest mapping, Jorum and HG are in this pattern. This is a clue / insight into how other glaciers that experience a sudden reduction in backstress at the grounding line might evolve in the future. Figure 2, Figure 7 – just a suggestion, make blue lines thicker, green lines thinner, to emphasize that the elevation change profile is the main point of the graphics. At first I thought the green line was binned elevation change rate (not area). Figure 3 – This graphic might be more effective as: (a) Speed, m/d, 2016; (b) Speed, 2016-2011; and Speed, 2016-1995 – same note for Figure 8. This would highlight the slowing in recent years. Also, check, is the date for the 1995 mapping November, as written, or October, as in the following figure profiles? Figure 4 and Figure 9 – Would it be possible to show the progression of speed versus time for the centerlines of the speed data – for example, in Figure 9, make the cross-section speed profile more narrow and place a center-line speed versus time graphic to the right of the plots? Similarly with Figure 4. Also – please place the location of the Fig9 flux gates and the Fig4 centerlines on one of the map views. I see that they are in the Supplemental Information, along with others, but it would be good to have these few in the main text maps to go with the figures.

---

## Short Comment (SC1) · 22 Jan 2018

T.S. Seehaus

thorsten.seehaus@fau.de

The manuscript provides a comprehensive and detailed analysis of the current glaciological changes along the north-eastern Antarctic Peninsula, a region which is highly dynamic and of high interest for the scientific community. The paper is well written and most of the methods are well described. However, the authors should provide some more detailed information on the analysis, in order to facilitate a better traceability of the results.

My main concern is about the horizontal and especially vertical registration of the TanDEM-X DEMs (L177)? No information is provided on this issue, which is an im-

portant processing step to generate accurate elevation change information.

Another mayor concern is about the deduction of the ice thickness along the flux gates. (L271) How is the ice thickness deduced for glaciers without sounding data and no floating terminus? What was done if sounding data was only available for sections of the flux gate profiles? Plots of the obtained ice thickness (indicating measured values and values obtained from the floatation criterion...) along the cross section profiles (e.g. in the supplement) would be helpful for the reader.

Kind Regards

Thorsten Seehaus

Some more minor issues regarding the description of the methods and discussion are listed below.

l167 please add a reference to other studies where this approach was already successfully applied to generate glacier mass balance information

l191: how is the RMSE calculated (mean value?, also not clearly mentioned in the supplement) and in the supplement L83: $S_e$ random error = RSME of SEC-IDHDT4 difference? It would be helpful to plot the coverage of the IDHDT4 data in one of the maps.

l199: how was the SAR backscatter intensity difference measured? Pixel by pixel or was any algorithm applied to reduce speckle noise?

l226: how was the bulk uncertainty estimated? Based on previous studies? In Section 2.1 the link to the respective section on error assessment in the supplement is missing and vice versa.

L252: where are the sample points located? Ice streams? Nunataks? Slow moving areas?

L361: what was the SMB under precollapse conditions? Did you try to calculate the ice

flux for precollapse conditions using your vel. field from 1993? How does it compare to the modeled results?

L452-459: Is there a correlation between the velocity of the detached ice blocks (or velocity relative to the glacier velocity) and the slowdown and frontal advance of the calving glaciers? This would be an interesting indicator, if there is any correlation.

L498: A discussion and explanation of the difference between your values and the findings by Scambos et al.(2014), would be helpful for the reader.

L530: Is there a correlation between acceleration events and retreat events and vice versa?

L537: Is there a reason, why the loss rate was higher in the period 2001-2008.

L552: What are other sources for the differences in sea ice pattern and ice drift?

Discussion: A more advanced discussion regarding the observed glacier changes in respect to changes in the local climate (e.g. considering recent publications on climatic conditions along the Peninsula: (Cook et al., 2016; Oliva et al., 2017; Turner et al., 2016) and previous mass balance estimates in this region (even just covering individual catchments) would be beneficial for the manuscript. Since, there is no mass balance information available for the whole study region before 2001 (by Scambos et al. 2014), a more detailed discussion of the temporal changes of the mass balance of at least some individual glaciers (where lit. data is available) would be quite interesting to illustrate the long-term evolution of the mass loss after the disintegration of the ice shelves in 1995 (PGC and Larsen A).

Table3: Is the calving flux the average interpolated value for the respective period?

Fig. 7 Caption: 2103 -> 2013

Supplement: Information on grounding line position in the plots in Section 1 (or magnified subsets) would be helpful in order to see which glaciers have floating sections

(Also in respect to the flux gate position)

Why are the tide model sample points in Fig. S3? Not explained in the paper and supplement?

---

## Referee Comment (RC2) · Anonymous Referee #2 · 2 Feb 2018

Review of "Changing pattern of ice flow and mass balance for glaciers discharging into the Larsen A and B embayments, Antarctic Peninsula, 2011 to 2016", by Rott et al.

This manuscript presents a mass balance study for the East of the northern Antarctic Peninsula, based on remotely sensed altitude changes, and a comparison with mass changes derived from modeled surface mass balance and surface flow velocities. It is shown that the land-based ice is continuing to lose mass, although the rate of loss has decreased over time since the collapse of the ice shelves. The manuscript is well written and clearly structured. The different data sources are described in great detail and also the methods are clearly presented. The results for the single catchment

areas are described and also provided as tables, which is quite useful. This is new data, continuing earlier observations to extend the overall record, and thus of great interest to the community. However , I would like to raise several points, which I think should be addressed before final publication:

1. in lines 272-276 error estimates for the glacier cross sections, the flow velocities and the modeled mass balance are given. They appear to come "out of the blue". It would be good to describe how you got to these estimates.

2. There is a general estimation of ice density to be 900 kg /mˆ3. This appears rather high to me, and means there is hardly any firn layer on the glaciers. There has to be an error assumption for this density value as well. This becomes important when you are concluding ice thickness from the floating glacier tongues, but also for the lowering of the surface of the grounded ice. Can there be a lowering of the surface due to firn compaction from surface melting? Maybe this could be discussed with the RACMO-results for the area. Is surface melting likely to occur on the glaciers?

3. The discussion is a bit disappointing in relation to the rest of the paper, and reads like an extended (repetitive) description of the results. I think one point which could be discussed more prominent in a broader context is the connection of mass loss and sea ice (fast ice) cover in the embayments. This could be an essential contribution to a better understanding of how ice shelves form, or the stability of existing ice shelves.

4. Figures 5 and 6: Maybe it would be better to place the labels of the catchment areas outside of them and connect with an error. In the form they are now they are obscuring quite a large part of the smaller catchments. The label B12 is missing, but as it's the only one maybe that's intentional.

---

## Author Comment (AC1) · 19 Feb 2018

**Response to "Changing pattern of ice flow and mass balance for glaciers discharging into the Larsen A and B embayments, Antarctic Peninsula, 2011 to 2016"**

The Cryosphere Discuss., **https://doi.org/10.5194/tc-2017-259**

*Authors*: Rott, H., Abdel Jaber, W., Wuite, J., Scheiblauer, S., Floricioiu, D., van Wessem, J.M., Nagler, T., Miranda, N., van den Broeke, M.R.

**Line numbers refer to the manuscript version (pdf) of 28 November 2017.**

*Referee comments in italic*

**Changes are tracked in the revised manuscript.**

**Anonymous Referee #1**

**Comment**: *The paper presents the results of a new analysis of elevation change and flow speed change for the eastern Antarctic Peninsula from Sjogren-Boydell glacier to Leppard Glacier, spanning the major outlets on the mainland Peninsula that were affected by the loss of ice shelves in 1995 and 2002. The study shows that the systems have moved in a positive direct in mass balance (either less negative, or positive outright) in the past few years. They attribute the decline of loss rate to the persistent presence of fast ice in the embayments.*

*This is a very clear and well-written study, with a lot of good (accurate) new data to offer. It could be published as it is. It provides a 'next chapter' in the monitoring of this rapidly-changing region impacted by ~25 years of very warm conditions (1980-2006) which have tapered to slightly cooling over the past several years (still warmer than the mid-20th century by a considerable amount). Even at this point, two decades past the ice shelf disintegrations, the glacier systems still show short-term changes in both elevation and flow speed.*

*Attribution of the reduction in ice losses (less negative mass balance, and in some cases a switch to positive mass balance) is given to a reduction in calving flux – i.e. a downstream movement of the calving front. This in turn is attributed to persistent fast ice.*

*Minor comments follow.*

**Response:** We are glad about the very positive feedback on the scope of our work and would like to thank the reviewer for the helpful comments, providing very valuable support for improving the quality of the manuscript. We carefully took into account these comments for revising the manuscript, as explained in the detailed response below.

**Comment:** *I would encourage the authors to adopt a consistent sign convention for mass budget / mass balance, i.e. negative means ice mass leaving the system. Although there is not a great deal of ambiguity (the words and numbers match the meaning everywhere I have checked), in places one wonders if a positive 'loss' might mean a net gain or a net loss, etc.*

**Response:** Wherever numbers on mass changes are shown, we now use consistently mass balance (negative for losses, positive for gain).

**Comment:** *L22 please use 'mass budget method' rather than 'input-output' method. Note that IMBIE-2 has now adopted this phrasing.*

**Response:** Everywhere changed.

**Comment:** *L32 no need to preface the mass budget results with 'Bn =' – it's a bit confusing, since you have not introduced that variable name, and not necessary.*

**Response:** Changed.

**Comment:** *L32 Also – L34 a 'mass loss' for a glacier losing mass would be –positive- : these are mass balance results, so negative numbers already mean 'loss'. A picky point, but this has been made strenuously by other authors/speakers.*

**Response:** Changed.

**Comment:** *L41 this 'sea ice cover' was/is 'landfast ice' – another picky point perhaps, but an upcoming paper will discuss this buttressing, and fast ice is a much better buttress than typical sea ice.*

**Response:** The proglacial fjords were actually filled by ice mélange extending several kilometres in front of the glaciers and by sea ice of different age farer away and also in the main sections of the Larsen A and B embayments. In the abstract we change the wording to "when in the proglacial fjords and bays ice mélange and sea ice persisted during summer." We address the topic of ice mélange in the revised discussion section.

**Comment:** *L68 here 'loss' is positive, as it should be with the phrasing; but better to stick to one numerical convention, positive or negative, and use words accordingly. It looks as though the majority of the Introduction uses positive numbers to report 'mass loss', and that is appropriate. However, it might be a bit confusing to people, since in terms of 'mass balance' these numbers should all be negative. In any case, please be consistent throughout the paper (abstract differs from main text). See L89-L100 and elsewhere.*

**Response:** Wherever numbers on mass changes are shown, we now refer consistently to mass balance (negative for losses, positive for gain).

**Comment:** *L76 change to '. . .began to accelerate and thin. . .' ('get thinner' is a bit colloquial, almost slang)*

**Response:** Changed.

**Comment:** *L154 'data takes' is also a bit colloquial; 'swath data' or 'data acquisitions'?*

**Response:** Date take is a technical term, referring to the specific ground coverage of SAR strip map mode images.

**Comment:** *L162, L165, L166 I think that "Raw DEMs" should be "raw DEMs" in English convention.*

**Response**: Raw DEM is a technical term (proper name) for a specific product within the DLR DEM processing chain.

**Comment:** *L173 change to '. . .data from the Antarctic Peninsula DEM. . ..'*

**Response:** Changed.

**Comment:** *L181 'swaths', not swathes (in US English at least)*

**Response:** Swathes is OK according to dictionary.

**Comment:** *L219 remove 'anywhere', and change to 'back-slope areas' - this is slightly confusing on first read.*

**Response:** Clarified, pointing out that this refers to any slope area.

**Comment:** *L254-255 – the RMSD is somewhat high, 50 – 60 m/yr, though, a bit of a concern.*

**Response:** Thanks for pointing this out. The comment makes clear that we did not well explain the uncertainties of the velocity data, which are specified in the details in the references cited on line 258 – 259 (same methods are used for work reported in this paper). In the supplementary material we added information on uncertainty estimate for the mass budget method (Section S3). The RMSD values between the TerraSAR-X and Sentinel-1 velocity products (line 254) can mainly be attributed to the different spatial resolution of the sensors. This is evident when considering the good agreement of mean velocities (0.011 m d$^{-1}$ and -0.002 m d$^{-1}$ for the two velocity components). The ice velocities for computing the calving fluxes are based on offset tracking using TerraSAR-X repeat pass SAR data. The uncertainty in the magnitude of the TerraSAR-X derived surface motion product at 50 m grid size is (conservatively) estimated at ±0.05 m d$^{-1}$ (details in Wuite et al., 2015). We compared also GPS and TerraSAR-X velocities at five stakes on Flask and Starbuck glaciers (with flow velocities < 1m d$^{-1}$); the differences of velocity magnitude between GPS and the satellite product are within ±0.01 m d$^{-1}$ (Supplementary Material).

**Comment:** *L260 again, please change to 'mass budget method'.*

**Response:** Changed.

**Comment:** *L297 '. . .approximately in a balanced state. . .' ; Same Note for L365. L311-312 see comment re Figures 3 and 8.*

**Response:** This statement is based on DEM differencing results, shown in Figure 1 and quantified in Table 1 for Larsen A, and in Figure 4 and Table 4 for Larsen B. We added references to the relevant figures and tables in the text. Figures 3 and 8 do not show mass balance.

**Comment:** *L510-513 – it would seem that several of the longer, thinner glaciers are evolving toward the Crane pattern of elevation change – the DBE system as well as Sjogren Glacier; and in the latest mapping, Jorum and HG are in this pattern. This is a clue / insight into how other glaciers that experience a sudden reduction in backstress at the grounding line might evolve in the future.*

**Response:** Thanks for pointing this out. However, in our view the future evolution of Crane Glacier retreat is unclear. The pattern of elevation change in recent years on Crane Glacier is different from that of other glaciers, showing a major shift of surface lowering up-glacier.

**Comment:** *Figure 2, Figure 7 – just a suggestion, make blue lines thicker, green lines thinner, to emphasize that the elevation change profile is the main point of the graphics. At first I thought the green line was binned elevation change rate (not area).*

**Response:** We changed the thickness of the lines as proposed.

**Comment:** *Figure 3 – This graphic might be more effective as: (a) Speed, m/d, 2016;(b) Speed, 2016-2011; and Speed, 2016-1995 – same note for Figure 8. This would highlight the slowing in recent years. Also, check, is the date for the 1995 mapping November, as written, or October, as in the following figure profiles?*

**Response:** Pronounced changes in velocity 2016 to 2011 affect only small sections of the velocity maps. We include insets in Figure 3 and Figure 8 focussing at the main glaciers.

**Comment:** *Figure 4 and Figure 9 – Would it be possible to show the progression of speed versus time for the centerlines of the speed data – for example, in Figure 9, make the cross-section speed profile more narrow and place a center-line speed versus time graphic to the right of the plots?*

**Response:** We added insets displaying time series of velocities for the centre of the flux gate in Figure 4. In Figure 9 the temporal sequence of gate velocities is quite evident.

**Comment:** *Similarly with Figure 4. Also – please place the location of the Fig9 flux gates and the Fig4 centerlines on one of the map views. I see that they are in the Supplemental Information, along with others, but it would be good to have these few in the main text maps to go with the figures.*

**Response:** We checked the feasibility of displaying the flux gates and centre lines in any of the figures of the main text (maps of elevation change and velocity). The lines do not show up clearly in these figures against the colour spectrum used for maps of velocity and elevation change. These lines show up much better in Figures S1 and S2, because the background image is bright (white/ light grey) and the maps are larger (one full page).

**Anonymous Referee #2**

**Comment:** *This manuscript presents a mass balance study for the East of the northern Antarctic Peninsula, based on remotely sensed altitude changes, and a comparison with mass changes derived from modeled surface mass balance and surface flow velocities. It is shown that the land-based ice is continuing to lose mass, although the rate of loss has decreased over time since the collapse of the ice shelves. The manuscript is well written and clearly structured. The different data sources are described in great detail and also the methods are clearly presented. The results for the single catchment areas are described and also provided as tables, which is quite useful. This is new data, continuing earlier observations to extend the overall record, and thus of great interest to the community. However, I would like to raise several points, which I think should be addressed before final publication:*

**Response:** The authors would like to thank the reviewer for the constructive comments, providing very valuable support for improving the quality of the manuscript. We took into account the issues raised in the comments for revising the manuscript, as explained in the response below.

**Comment 1.** *In lines 272-276 error estimates for the glacier cross sections, the flow velocities and the modelled mass balance are given. They appear to come "out of the blue". It would be good to describe how you got to these estimates.*

Details on the uncertainty estimate for mass balance at basin scale, derived by means of the mass budget method are specified in the revised version of the Supplementary Material, Section 3.2, accounting for uncertainties in flow velocity and ice thickness at the flux gates and for uncertainties of surface mass balance (SMB) and

**Comment 2**. *There is a general estimation of ice density to be 900 kg /$m^3$. This appears rather high to me, and means there is hardly any firn layer on the glaciers. There has to be an error assumption for this density value as well. This becomes important when you are concluding ice thickness from the floating glacier tongues, but also for the lowering of the surface of the grounded ice. Can there be a lowering of the surface due to firn compaction from surface melting? Maybe this could be discussed with the RACMO-results for the area. Is surface melting likely to occur on the glaciers?*

**Response:** For converting volume change over a certain time span to the change in mass an estimate for the change in density of the vertical snow and ice column is needed. If the mean density of the column remains unchanged the water equivalent can be obtained by multiplying the elevation change by the density of ice. Changes in density and microstructure of the firn volume would cause changes in the X-band radar backscatter coefficient. From the similarity of the radar backscatter coefficients in the 2011 and 2016 TanDEM-X images we can exclude significant changes in the structure and mean density of the snow/firn column. Furthermore, the good agreement between the IceBridge lidar based dh/dt values and the TanDEM-X based dh/dt values indicates also stability of the density and vertical structure of the snow/ice medium. Changes of density and structure would cause a vertical shift of the radar scattering phase centre within the volume, resulting in a relative shift of the surface in the SAR DEM versus the surface in the optical data (Dall, 2007). These two features indicate that the possible error due to change of mean density of the vertical column is negligible compared to

the uncertainty in dh/dt. We use a mean density of 900 kg m$^{-3}$ which is commonly used for geodetic mass balance studies (e.g. Cogley, 2009; Haug et al., 2009). Scambos et al. (2014) use also a mean density of 900 kg m$^{-3}$ for converting volume change into mass for their mass balance analysis of glaciers on the northern Antarctic Peninsula over the period 2001 to 2010, so that these data can be directly compared with our estimates.

The radar backscatter coefficients (sigma-0) and structural patterns at the flux gates in the TerraSAR-X images correspond to typical features of glacier ice so that the density of ice can be used for computing the calving fluxes. The surface patterns and co-polarized sigma-0 values (-6 dB to -9 dB) in the transition zones from grounded ice to floating ice of SCAR inlet ice shelf are indicating blue ice. Frozen firn further upstream has typical sigma-0 values between -1 dB and -3 dB. The ice at the fronts of the calving glaciers is highly fractured so that corner effects cause higher sigma-0 values than those of blue ice, but the structural patterns is typical for exposed glacier ice. The Landsat image of 29 October 2016 shows reduced surface reflectivity for the lower terminus section of the calving glaciers, also an indication for exposed glacier ice.

**Comment** 3. *The discussion is a bit disappointing in relation to the rest of the paper, and reads like an extended (repetitive) description of the results. I think one point which could be discussed more prominent in a broader context is the connection of mass loss and sea ice (fast ice) cover in the embayments. This could be an essential contribution to a better understanding of how ice shelves form, or the stability of existing ice shelves.*

**Response:** Thanks for pointing this out and the suggestions. We add more substance to the discussion by referring to studies on the buttressing effects of ice mélange for calving behaviour and discussing possible impacts of recent changes in climate conditions and atmospheric circulation.

In 2013 to 2016 the fjords and bays in front of the glaciers were covered by ice mélange, whereas the wider area of the Larsen A and B embayments was covered by sea ice of different age. To illustrate these features we include in the revised Complementary Material a TerraSAR-X image covering the proglacial fjord of Crane, Jorum and Punchbowl glaciers (Figure S4). In the discussion we refer to publications on models and observations of the buttressing effect by ice mélange affecting the seasonal advance and retreat of calving fronts of Greenland outlet glaciers (Walter et al., 2012; Todd and Christofferson, 2014; Amundson et al., 2016). Major advancements on the role of proglacial ice mélange and persistent sea ice cover for ice shelf formation would require substantial modelling work which is beyond the scope of this paper.

**Comment** 4: *Figures 5 and 6: Maybe it would be better to place the labels of the catchment areas outside of them and connect with an arrow. In the form they are now they are obscuring quite a large part of the smaller catchments. The label B12 is missing, but as it's the only one maybe that's intentional.*

**Response:** We modified Figures 5 and 6 as suggested.

**Comments by T.S. Seehaus**

**Comment:** *The manuscript provides a comprehensive and detailed analysis of the current glaciological changes along the north-eastern Antarctic Peninsula, a region which is highly dynamic and of high interest for the scientific community. The paper is well written and most of the methods are well described. However, the authors should provide some more detailed information on the analysis, in order to facilitate a better traceability of the results.*

*My main concern is about the horizontal and especially vertical registration of the TanDEM-X DEMs (L177)? No information is provided on this issue, which is an important processing step to generate accurate elevation change information. Another mayor concern is about the deduction of the ice thickness along the flux gates. (L271) How is the ice thickness deduced for glaciers without sounding data and no floating terminus? What was done if sounding data was only available for sections of the flux gate profiles? Plots of the obtained ice thickness (indicating measured values and values obtained from the floatation criterion...) along the cross section profiles (e.g. in the supplement) would be helpful for the reader.*

*Some more minor issues regarding the description of the methods and discussion are listed below.*

**Response:** We wish to thank Thorsten Seehaus for his comments. We respond first to the critical issues raised above. The response to the minor issues is given below, point by point.

*Main concern is about the horizontal and especially vertical registration of the TanDEM-X DEMs*: This is a flat statement without providing any information why this should be of concern. The excellent agreement with the IceBridge ATM dh/dt data would be impossible without precise geocoding and horizontal/vertical registration of the TanDEM-X DEMs. Results of this comparison and of the uncertainty analysis are detailed in Section 3 of the Supplementary Material (probably overlooked). In order to further underline the good performance of the dh/dt analysis we add in the main manuscript a figure showing a detailed comparison of dh/dt for the ATM transect on Crane Glacier (new Figure 1). The excellent agreement (coefficient of determination $R^2 = 0.98$) confirms the high quality of both data sets which were independently processed. Processing of the TanDEM-X dh/dt maps was completed before the ATM dh/dt data became available at the IceBridge web site. The main processing steps are described on page 6 of the manuscript; technical procedures are described in detail in the references.

*How is the ice thickness deduced for glaciers without sounding data and no floating terminus?* There are no glaciers without either sounding or non-floating terminus among the glaciers for which the mass budget method (IOM) was applied (which is a subset of the glaciers analysed by DEM differencing). The same gates are used as in previous studies, details are described there, and for some of the glaciers cross sections are shown (Rott et al., 2001; Wuite et al., 2015).

**Comment:** *l167 please add a reference to other studies where this approach was already successfully applied to generate glacier mass balance information.*

**Response:** The paper includes already four pages of references, including several papers on mass balance studies that are of immediate relevance to the work reported here. This is not a review paper, nor a technical paper on mass balance methods.

**Comment:** l191: *how is the RMSE calculated (mean value?, also not clearly mentioned in the supplement) and in the supplement L83: S_e random error = RSME of SEC-IDHDT4 difference? It would be helpful to plot the coverage of the IDHDT4 data in one of the maps.*

**Response:** The uncertainty estimate is described in detail in Section 3 of Supplementary Material. Nevertheless we repeat some of the main points here. The new Figure 1 (see below) may also contribute to better understanding. The pixel by pixel comparison of ATM dh/dt cells (250 m x 250 m) and co-located TDM dh/dt data (7 x 7 pixels average) yields RMSD values between 0.15 m a$^{-1}$ and 0.35 m a$^{-1}$ for the six galciers (Supplement Table S3). Some of this difference is due to the time shift of the two date sets. We use RMSD = 0.28 m a$^{-1}$ as a representative value and assume that the differences result from uncertainties in both data sets. This results in RMSE = 0.20 m a$^{-1}$ at the spatial scale of the ATM pixels, derived from the five year dh/dt analysis. For the random error in dh/dt maps (Line S79, first term under the square root and specs in line S83) we account for the shorter time interval, increasing the value RMSE = 0.20 m a$^{-1}$ from the 5-year time span to 0.39 m a$^{-1}$ (3 years) and 0.58 m a$^{-1}$ (2 years). The independent number of samples is based on a distance for spatial decorrelation of 500 m (a rather conservative estimate for a high resolution DEM). All this information (and more) is provided already in Section 3 of Supplementary Material.

The IDHDT4 data (and plots of the locations) used for the intercomparison on the six glaciers (specified in Table S3) are available at the IceBridge web site (see web link in Studinger, 2017). The plots shown in the manuscript are already overloaded with information. We recommend looking at the original data and maps which show the IceBridge flight tracks and report also technical details, performance, etc. of the IDHDT4 products.

**Comment:** *l199: how was the SAR backscatter intensity difference measured? Pixel by pixel or was any algorithm applied to reduce speckle noise?*

**Response:** Speckle filtering should not be applied for deriving backscatter coefficients from SAR images because speckle filters change the statistics of the return signals and consequently the radiometry (intensity). We apply averaging in the linear (non-logarithmic) scale to obtain SAR backscatter intensity.

**Comment:** *l226: how was the bulk uncertainty estimated? Based on previous studies? In Section 2.1 the link to the respective section on error assessment in the supplement is missing and vice versa.*

**Response:** This estimate is also based on the intercomparison of the ATM and TanDEM-X DEM data, allowing for an additional margin. Because of the availability of these two coincident and independent data sets it would not be meaningful using uncertainty estimates from other studies. Additional information is provided in the revised version of Supplementary Material.

**Comment:** *L252: where are the sample points located? Ice streams? Nunataks? Slow moving areas?*

**Response:** The comparison includes all areas where the velocity maps of the two sensors are overlapping**.**

**Comment:** *L361: what was the SMB under precollapse conditions? Did you try to calculate the ice flux for precollapse conditions using your vel. field from 1993? How does it compare to the modeled results?*

**Response:** The pre-collapse SMB was only slightly lower ($b_n$ = 1268 kg m$^{-2}$ a$^{-1}$ for 1990 to 1995) than the 2011-2016 SMB. A problem for retrieving the pre-collapse calving flux are missing proper data for the ice thickness at the gate. The modelled ice thickness by Huss and Farinotti (2014) seems to underestimate the ice thickness at the calving gate significantly. The thickness inferred by the floatation criterion in the 2011 to 2016 time frame exceeds the Huss and Farinotti estimate, in spite of major surface lowering since the collapse.

**Comment***: L452-459: Is there a correlation between the velocity of the detached ice blocks (or velocity relative to the glacier velocity) and the slowdown and frontal advance of the calving glaciers? This would be an interesting indicator, if there is any correlation.*

**Response:** We did not perform a detailed analysis on this. A detailed study on motion fields of ice mélange and sea ice in the proglacial fjords and their temporal evolution would certainly be of interest, the topic for future studies.

**Comment***: L498: A discussion and explanation of the difference between your values and the findings by Scambos et al.(2014), would be helpful for the reader.*

**Response:** This is already discussed by Wuite et al. (2015). We briefly take this up again in the discussion section.

**Comment:** *L530: Is there a correlation between acceleration events and retreat events and vice versa?*

**Response:** We did not check**.** A correlation of these events per se would not add much to understanding.

**Comment:** *L537: Is there a reason, why the loss rate was higher in the period 2001-2008.*

**Response:** This can be attributed to decrease in velocities after 2008. This is reported in detail in several of the papers cited in the introduction and discussion sections (e.g. for Larsen B in Wuite et al., 2015).

**Comment:** *L552: What are other sources for the differences in sea ice pattern and ice drift?*

**Response:** We checked time series of SAR images (ERS SAR, Envisat ASAR, TerraSAR-X, Sentinel-1) for coverage by sea ice or open water in the Larsen A and B embayments. A detailed analysis of sea ice motion fields and their temporal evolution would be a topic of great interest, however beyond the scope of this paper.

**Comment:** *Discussion: A more advanced discussion regarding the observed glacier changes in respect to changes in the local climate (e.g. considering recent publications on climatic conditions along the Peninsula: (Cook et al., 2016; Oliva et al., 2017; Turner et al.,2016) and previous mass balance estimates in this region (even just covering individual catchments) would be beneficial for the manuscript. Since, there is no mass balance information available*

*for the whole study region before 2001 (by Scambos et al. 2014), a more detailed discussion of the temporal changes of the mass balance of at least some individual glaciers (where lit. data is available) would be quite interesting to illustrate the long-term evolution of the mass loss after the disintegration of the ice shelves in 1995 (PGC and Larsen A).*

**Response:** Thanks for these suggestions. Various papers cited in the introduction and discussion deal with changes on mass balance and climate in this region. Oliva et al. (2017) and Turner et al. (2016) are already cited in the first version of the manuscript. Our focus is on the period 2011 to 2016 and the paper is already quite lengthy. We expanded the discussion on topics suggested by Referee 2.

**Comment:** *Table3: Is the calving flux the average interpolated value for the respective period?*

**Response:** Yes.

**Comment:** *Fig. 7 Caption: 2103 -> 2013*

**Response:** Corrected

**Comment:** *Supplement: Information on grounding line position in the plots in Section 1 (or magnified subsets) would be helpful in order to see which glaciers have floating sections (Also in respect to the flux gate position)*

**Response:** In 2011 the floating glacier sections were quite small (except on DBE glaciers), not far inland of the 2011 glacier front. It would be difficult to discriminate these lines in the maps shown in the manuscript and supplement. Detailed maps and time series at the scale of individual glaciers would be needed, a topic for future papers.

**Comment:** *Why are the tide model sample points in Fig. S3? Not explained in the paper and supplement?*

**Response:** Thanks for noting this. The background map (with coastlines, lat/lon markers, etc.) was generated for another project; here we use it to show the coverage of the TDM DEMs. We did not use any tide model for the dh/dt analysis. Corrected in the revised version.

**References:**

Amundson, J. M., Fahnestock, M., Truffer, M., Brown, J., Lüthi, M.P., and Motyka, R.J.: Ice mélange dynamics and implications for terminus stability, Jakobshavn Isbræ, Greenland, J. Geophys. Res., 115, F01005, doi:10.1029/2009JF001405, 2016.

Cogley, J. G.: Geodetic and direct mass-balance measurements: comparison and joint analysis, Ann. Glaciol., 50, 96-100, 2009.

Dall, J.: InSAR elevation bias caused by penetration into uniform volumes. IEEE Trans. Geosc. Remote Sensing, 45(7), 2319–2324, 2007.

Haug, T., Rolstad, C., Elvehøy, H., Jackson, M., and I. Maalen-Johansen: (2009), Geodetic mass balance of the western Svartisen ice cap, Norway, in the periods 1968–1985 and 1985–2002, Ann. Glaciol., 50, 119-125, 2009.

Huss, M. and Farinotti, D.: A high-resolution bedrock map for the Antarctic Peninsula, The Cryosphere, 8, 1261–1273,doi:10.5194/tc-8-1261-2014, 2014.

Rott, H., Müller, F., Nagler, T., and Floricioiu, D.: The imbalance of glaciers after disintegration of Larsen B Ice Shelf, Antarctic Peninsula, The Cryosphere, 5 (1), 125–134, doi:10.5194/tc-5-125-2011, 2011.

Rott, H., Floricioiu, D., Wuite, J., Scheiblauer, S., Nagler, T., and Kern, M.: Mass changes of outlet glaciers along the Nordensjköld Coast, northern Antarctic Peninsula, based on TanDEM-X satellite measurements, Geophys. Res. Lett., 41, doi:10.1002/2014GL061613, 2014.

Scambos, T. A., Berthier, E., Haran, T., Shuman, C. A., Cook, A. J., Ligtenberg, S. R. M., and Bohlander, J.: Detailed ice loss pattern in the northern Antarctic Peninsula: widespread decline driven by ice front retreats, The Cryosphere, 8, 2135-2145, doi:10.5194/tc-8-2135-2014, 2014.

Studinger, M. S: IceBridge ATM L4 Surface Elevation Rate of Change, Version 1, Subset M699, S10, Boulder, Colorado USA. NASA National Snow and Ice Data Center Distributed Active Archive Center, doi: http://dx.doi.org/10.5067/BCW6CI3TXOCY, 2017.

Todd, J. and Christoffersen, P.: Are seasonal calving dynamics forced by buttressing from ice mélange or undercutting by melting? Outcomes from full-Stokes simulations of Store Glacier, West Greenland, The Cryosphere, 8, 2353-2365, https://doi.org/10.5194/tc-8-2353-2014, 2014.

**Additional figure for manuscript**

[Figure]

**Figure 1**. Scatterplot of measurements of surface elevation change (dh/dt) 2016 - 2011 along the central flowline of Crane Glacier based on IceBrigde ATM and TanDEM-X elevation data. The line shows the linear fit.

---

## Author Comment (AC3) · 19 Feb 2018

*Supplement of*

**Changing pattern of ice flow and mass balance for glaciers discharging into the Larsen A and B embayments, Antarctic Peninsula, 2011 to 2016**

H. Rott et al.

*Correspondence to*: Helmut.Rott@enveo.at

**Contents**

**Section S1 – Overview on glacier basins for retrieval of volume change and mass balance**

The outlines of the glacier basins for retrieval of volume change and mass balance are displayed on a Landsat image (Figures S1 and S2). Table S1 contains a list of the basins with area extent in 2013 and 2016 and the GLIMS ID for the main glacier in each basin.

**Section S2 – Data coverage by TanDEM-X interferometric SAR data**

A map with area coverage of the TanDEM-X SAR image tracks used for DEM retrieval is shown (Fig. S3) and the specifications of the DEMs used for generating surface elevation change (SEC) products are listed (Table S2).

**Section S3 – Estimation of uncertainty for surface elevation change and mass budget**

Details are presented on the procedures and data base for estimating the uncertainty of the TanDEM-X maps of surface elevation change and on the mass balance obtained by DEM differencing and by the mass budget method.

**Section S4 – TerraSAR-X image with ice mélange and sea ice in proglacial fjord**

A TerraSAR-X image is shown in order to illustrate typical patterns of ice mélange and sea ice in proglacial fjords during years of persistent sea ice cover.

[revised manuscript text omitted]
. The bias estimates for the ice plateau and the unsurveyed slopes are also deduced from the comparison of ATM and TDM dh/dt data. They are based on the first two terms under the square root of Eq. S1, applying $f_m = 1.0$, n = 80, and increasing the resulting value by 50 % to allow for an additional margin.

For converting volume change to mass change we assume a mean density $\rho = 900\ kg\ m^{-3}$.

This value is commonly used for geodetic mass balance studies in case the mean density of the snow/ice column does not change. From the similarity of the radar backscatter coefficients in the 2011 and 2016 TanDEM-X images we can exclude significant changes in the structure and density of the snow/firn column. The good agreement between the IceBridge lidar based dh/dt values and the TanDEM-X based dh/dt values indicates also stability of the structure and density of the snow/ice medium. Changes of density and structure would cause a vertical shift of the radar scattering phase centre within the volume, resulting in a relative shift of the surface in the SAR DEM data versus the surface in the optical data (Dall, 2007). The stability of radar backscatter and the good agreement of optical and radar dh/dt indicate that the possible error due to density changes in the vertical column is negligible compared to the uncertainty in dh/dt. Scambos et al. (2014) use also a mean density of 900 kg m$^{-3}$ for converting volume change into mass for their mass balance analysis of glaciers on the northern Antarctic Peninsula over the period 2001 to 2010, so that these mass changes can be directly compared with our estimates.

For estimating the uncertainty of sub-regions (Larsen A, Larsen B embayment, SCAR Inlet) we assume that the errors for glaciers covered by a single TDM track are correlated (the errors are added) and the errors of different tracks are uncorrelated.

**3.2 Mass budget method**

The uncertainty estimate for mass balance at basin scale, derived by means of the mass budget method, accounts for uncertainties of surface mass balance (SMB) and for uncertainties in flow velocity and ice thickness at the flux gates. The SMB is based on output of the regional climate model RACMO Version 2.3p2 (van Wessem et al., 2016; 2017). For the uncertainty of surface mass balance at basin scale we assume ± 15 % uncertainty for the average SMB. This value is based on an evaluation of the RACMO SMB output on the Antarctic Peninsula, showing good agreement with the balance flux of Larsen B glaciers in pre-collapse state (van Wessem et al., 2016).

The velocities used for computing calving fluxes are exclusively derived from TerraSAR-X and TanDEM-X repeat pass data. The uncertainty in the magnitude of the TerraSAR-X derived surface motion product over Larsen B glaciers at 50 m grid size is estimated at ±0.05 m d$^{-1}$ (details in Wuite et al., 2015). Besides, we performed a direct comparison between GPS velocity measurements at stakes made by British Antarctic Survey (BAS) over annual intervals 2009/2010 and 2011/2012 on Flask Glacier and 2011/2012 on Starbuck Glacier. (GPS data kindly made available by BAS for the ESA project GlacAPI: https://glacapi.enveo.at/). Although the TerraSAR-X measurements cover shorter intervals, the agreement is very good: The differences between GPS and TerraSAR-X velocities at the individual stakes are within ±0.01 m d$^{-1}$; the stake velocities ranging from 0.71 to 0.95 m d$^{-1}$ on Flask Glacier and 0.13 to 0.18 m d$^{-1}$ on Starbuck Glacier. This confirms that an uncertainty estimate of ±0.05 m d$^{-1}$ at 50 m grid size is a rather conservative estimate for the TerraSAR-X/ TanDEM-X based velocity product. For computing calving fluxes we assume ± 5 % uncertainty in velocity across the gate. For uncertainty estimates of mass fluxes we assume ± 10 % error for the cross section area of glaciers with GPR data across or close to the gates and

± 15 % for glaciers where the ice thickness is deduced from frontal height above flotation
measured by TanDEM-X (relative 90 % point-to-point height error <2 m; Rizzoli et al., 2012).

**Section 4 – TerraSAR-X image with ice mélange and sea ice in proglacial fjord**

Figure 4 shows typical patterns of ice mélange (a mixture of icebergs and bergy bits, held
together by sea ice) in the fjord in front of Crane, Jorum and Punchbowl glaciers, and sea ice
with a larger iceberg in the wider section of the bay.

[Figure]

**Figure S4.** Section of TerraSAR-X amplitude image, acquired on 27 July 2016, with calving
fronts of Crane ( C), Jorum (J) and Punchbowl (P) glaciers and the proglacial fjord covered by
ice mélange and sea ice.

---

## Author Response (AR1)

**Response to "Changing pattern of ice flow and mass balance for glaciers discharging into the Larsen A and B embayments, Antarctic Peninsula, 2011 to 2016"**

The Cryosphere Discuss., **https://doi.org/10.5194/tc-2017-259**

*Authors*: Rott, H., Abdel Jaber, W., Wuite, J., Scheiblauer, S., Floricioiu, D., van Wessem, J.M., Nagler, T., Miranda, N., van den Broeke, M.R.

**Line numbers in the response refer to the revised version with changes tracked, dated 2018-03-07**

**In the revised manuscript the figure numbers are shifted by 1 because an additional figure (new Figure 1) was added.** This figure, demonstrating excellent agreement between elevation change measured by IceBridge airborne lidar and TanDEM-X, responds to a critical question of Interactive Comment SC1 regarding the quality of the TanDEM-X DEM differencing analysis.

*Referee comments in italic*

**Anonymous Referee #1**

**Comment**: *The paper presents the results of a new analysis of elevation change and flow speed change for the eastern Antarctic Peninsula from Sjogren-Boydell glacier to Leppard Glacier, spanning the major outlets on the mainland Peninsula that were affected by the loss of ice shelves in 1995 and 2002. The study shows that the systems have moved in a positive direct in mass balance (either less negative, or positive outright) in the past few years. They attribute the decline of loss rate to the persistent presence of fast ice in the embayments.*

*This is a very clear and well-written study, with a lot of good (accurate) new data to offer. It could be published as it is. It provides a 'next chapter' in the monitoring of this rapidly-changing region impacted by ~25 years of very warm conditions (1980-2006) which have tapered to slightly cooling over the past several years (still warmer than the mid-20th century by a considerable amount). Even at this point, two decades past the ice shelf disintegrations, the glacier systems still show short-term changes in both elevation and flow speed.*

*Attribution of the reduction in ice losses (less negative mass balance, and in some cases a switch to positive mass balance) is given to a reduction in calving flux – i.e. a downstream movement of the calving front. This in turn is attributed to persistent fast ice.*

*Minor comments follow.*

**Response:** We are glad about the very positive feedback on the scope of our work and would like to thank the reviewer for the helpful comments, providing very valuable support for improving the quality of the manuscript. We carefully took into account these comments for revising the manuscript, as explained in the detailed response below.

**Comment:** *I would encourage the authors to adopt a consistent sign convention for mass budget / mass balance, i.e. negative means ice mass leaving the system. Although there is not*

*a great deal of ambiguity (the words and numbers match the meaning everywhere I have checked), in places one wonders if a positive 'loss' might mean a net gain or a net loss, etc.*

**Response:** We now use consistently mass balance (negative for losses, positive for gain).

**Changes in manuscript:** This sign convention is now used throughout the manuscript.

**Comment:** *L22 please use 'mass budget method' rather than 'input-output' method. Note that IMBIE-2 has now adopted this phrasing.*

**Changes in manuscript:** Everywhere changed as proposed.

**Comment:** *L32 no need to preface the mass budget results with 'Bn =' – it's a bit confusing, since you have not introduced that variable name, and not necessary.*

**Changes in manuscript:** Changed. $B_n$ deleted in the Abstract.

**Comment:** *L32 Also – L34 a 'mass loss' for a glacier losing mass would be –positive- : these are mass balance results, so negative numbers already mean 'loss'. A picky point, but this has been made strenuously by other authors/speakers.*

**Changes in manuscript:** Changed. Mass balance numbers are now used throughout the abstract.

**Comment:** *L41 this 'sea ice cover' was/is 'landfast ice' – another picky point perhaps, but an upcoming paper will discuss this buttressing, and fast ice is a much better buttress than typical sea ice.*

**Response:** The proglacial fjords were actually filled by ice mélange extending several kilometres in front of the glaciers and by sea ice of different age farer away and also in the main sections of the Larsen A and B embayments.

**Changes in manuscript:** In L42, L43 (revised manuscript) we change the wording to "when ice mélange and sea ice cover persisted in the proglacial fjords and bays during summer." We address the topic of ice mélange in the revised discussion section.

**Comment:** *L68 here 'loss' is positive, as it should be with the phrasing; but better to stick to one numerical convention, positive or negative, and use words accordingly. It looks as though the majority of the Introduction uses positive numbers to report 'mass loss', and that is appropriate. However, it might be a bit confusing to people, since in terms of 'mass balance' these numbers should all be negative. In any case, please be consistent throughout the paper (abstract differs from main text). See L89-L100 and elsewhere.*

**Changes in manuscript:** Wherever numbers on mass changes are shown, we now refer consistently to mass balance (negative for losses, positive for gain).

**Comment:** *L76 change to '. . .began to accelerate and thin. . .' ('get thinner' is a bit colloquial, almost slang)*

**Changes in manuscript:** Wording **c**hanged as proposed.

**Comment:** *L154 'data takes' is also a bit colloquial; 'swath data' or 'data acquisitions'?*

**Response:** Date take is a technical term, referring to the specific ground coverage of SAR strip map mode images. Therefore we keep this term.

**Comment:** *L162, L165, L166 I think that "Raw DEMs" should be "raw DEMs" in English convention.*

**Response**: Raw DEM is a technical term (proper name) for a specific product within the DLR DEM processing chain. Therefore we keep this term.

**Comment:** *L173 change to '. . .data from the Antarctic Peninsula DEM. . ..'*

**Changes in manuscript:** Wording changed as proposed.

**Comment:** *L181 'swaths', not swathes (in US English at least)*

**Response:** Swathes is OK according to dictionary.

**Comment:** *L219 remove 'anywhere', and change to 'back-slope areas' - this is slightly confusing on first read.*

**Changes in manuscript:** Clarified, pointing out that this refers to any slope area (L229 of revised manuscript).

**Comment:** *L254-255 – the RMSD is somewhat high, 50 – 60 m/yr, though, a bit of a concern.*

**Response:** Thanks for pointing this out. The comment makes clear that we did not well explain the uncertainties of the velocity data, which are specified in the details in the references cited on line 258 – 259 (same methods are used for work reported in this paper). In the Supplementary Material we added information on uncertainty estimate for the mass budget method (Section S3). The RMSD values between the TerraSAR-X and Sentinel-1 velocity products (line 254) can mainly be attributed to the different spatial resolution of the sensors. This is evident when considering the good agreement of mean velocities (0.011 m d$^{-1}$ and -0.002 m d$^{-1}$ for the two velocity components). The ice velocities for computing the calving fluxes are based on offset tracking using TerraSAR-X repeat pass SAR data. The uncertainty in the magnitude of the TerraSAR-X derived surface motion product at 50 m grid size is (conservatively) estimated at ±0.05 m d$^{-1}$ (details in Wuite et al., 2015). We compared also GPS and TerraSAR-X velocities at five stakes on Flask and Starbuck glaciers (with flow velocities < 1m d$^{-1}$); the differences of velocity magnitude between GPS and the satellite product are within ±0.01 m d$^{-1}$ (Supplementary Material).

**Changes in manuscript:** We explain that the RMSD values of TerraSAR-X and Sentinel-1 velocities are mainly due to the different spatial resolutions (L267, L268 in revised manuscript) and that velocities at flux gates are derived from TerraSAR-X images (L291, L292). The uncertainty of TerraSAR-X is specified in L254, L255.

**Comment:** *L260 again, please change to 'mass budget method'.*

**Changes in manuscript:** Changed as proposed.

**Comment:** *L297 '. . .approximately in a balanced state. . .' ; Same Note for L365. L311-312 see comment re Figures 3 and 8.*

**Response:** This statement is based on DEM differencing results, shown in the dh/dt maps and quantified in Table 1 for Larsen A and in Table 4 for Larsen B. Figures 3 and 8 (Fig. numbers of 1$^{st}$ manuscript version) are velocity maps and do not show mass balance.

**Changes in manuscript:** We added references to the relevant figures and tables in the text to make this clear (L316 for Larsen A, L402 for Larsen B).

**Comment:** *L510-513 – it would seem that several of the longer, thinner glaciers are evolving toward the Crane pattern of elevation change – the DBE system as well as Sjogren Glacier; and in the latest mapping, Jorum and HG are in this pattern. This is a clue / insight into how other glaciers that experience a sudden reduction in backstress at the grounding line might evolve in the future.*

**Response:** Thanks for pointing this out. We addressed this in L581 to L583 of the 1[st] manuscript version.

**Comment:** *Figure 2, Figure 7 – just a suggestion, make blue lines thicker, green lines thinner, to emphasize that the elevation change profile is the main point of the graphics. At first I thought the green line was binned elevation change rate (not area).*

**Changes in manuscript:** We changed the thickness of the lines as proposed.

**Comment:** *Figure 3 – This graphic might be more effective as: (a) Speed, m/d, 2016;(b) Speed, 2016-2011; and Speed, 2016-1995 – same note for Figure 8. This would highlight the slowing in recent years. Also, check, is the date for the 1995 mapping November, as written, or October, as in the following figure profiles?*

**Changes in manuscript:** Pronounced changes in velocity 2016 to 2011 affect only small sections of the velocity maps. We include insets in Figure 4 and Figure 9 for the main glaciers.

**Comment:** *Figure 4 and Figure 9 – Would it be possible to show the progression of speed versus time for the centerlines of the speed data – for example, in Figure 9, make the cross-section speed profile more narrow and place a center-line speed versus time graphic to the right of the plots?*

**Changes in manuscript:** We added insets displaying time series of velocities for the centre of the flux gate in Figure 5 (previous Fig. 4). In Figure 10 (previous Fig. 9) we changed the colour code so that the temporal sequence of gate velocities is clearer.

**Comment:** *Similarly with Figure 4. Also – please place the location of the Fig9 flux gates and the Fig4 centerlines on one of the map views. I see that they are in the Supplemental Information, along with others, but it would be good to have these few in the main text maps to go with the figures.*

**Response:** We checked the feasibility of displaying the flux gates and centre lines in any of the figures of the main text (maps of elevation change and velocity). The lines do not show up clearly in these figures against the colour spectrum used for velocity and elevation change. These lines show up much better in Figures S1 and S2, because the background image is bright (white/ light grey) and the maps are larger (one full page).

**Anonymous Referee #2**

**Comment:** *This manuscript presents a mass balance study for the East of the northern Antarctic Peninsula, based on remotely sensed altitude changes, and a comparison with mass changes derived from modeled surface mass balance and surface flow velocities. It is shown*

*that the land-based ice is continuing to lose mass, although the rate of loss has decreased over time since the collapse of the ice shelves. The manuscript is well written and clearly structured. The different data sources are described in great detail and also the methods are clearly presented. The results for the single catchment areas are described and also provided as tables, which is quite useful. This is new data, continuing earlier observations to extend the overall record, and thus of great interest to the community. However, I would like to raise several points, which I think should be addressed before final publication:*

**Response:** The authors would like to thank the reviewer for the constructive comments, providing very valuable support for improving the quality of the manuscript. We took into account the issues raised in the comments for revising the manuscript, as explained in the response below.

**Comment 1.** *In lines 272-276 error estimates for the glacier cross sections, the flow velocities and the modelled mass balance are given. They appear to come "out of the blue". It would be good to describe how you got to these estimates.*

**Response:** We agree that error estimates for the mass budget method were not well explained. In the revised version of the Supplementary Material we include a new section (3.2) explaining uncertainties in flow velocity and ice thickness at the flux gates and uncertainties of surface mass balance (SMB). In the revised manuscript we add part of this information and make reference to the section in Supplementary Material.

**Changes in manuscript:** In L286 to L288 reference is made new section 3.2 in the Supplementary Material with uncertainty analysis for the mass budget method. Furthermore, in the revised manuscript we make clear that velocities at flux gates are exclusively derived from TerraSAR-X images (L291, L292). The uncertainty of TerraSAR-X is specified in L254, L255).

**Comment 2**. *There is a general estimation of ice density to be 900 kg /$m^3$. This appears rather high to me, and means there is hardly any firn layer on the glaciers. There has to be an error assumption for this density value as well. This becomes important when you are concluding ice thickness from the floating glacier tongues, but also for the lowering of the surface of the grounded ice. Can there be a lowering of the surface due to firn compaction from surface melting? Maybe this could be discussed with the RACMO-results for the area. Is surface melting likely to occur on the glaciers?*

**Response:** For converting volume change over a certain time span to the change in mass an estimate for the change in density of the vertical snow and ice column is needed. If the mean density of the column remains unchanged the water equivalent can be obtained by multiplying the elevation change by the density of ice. Changes in density and microstructure of the firn volume would cause changes in the X-band radar backscatter coefficient. From the similarity of the radar backscatter coefficients in the 2011 and 2016 TanDEM-X images we can exclude significant changes in the structure and mean density of the snow/firn column. Furthermore, the good agreement between the IceBridge lidar based dh/dt values and the TanDEM-X based dh/dt values indicates also stability of the density and vertical structure of the snow/ice medium. Changes of density and structure would cause a vertical shift of the radar scattering phase centre within the volume, resulting in a relative shift of the surface in the SAR DEM

versus the surface in the optical data (Dall, 2007). These two features indicate that the possible error due to change of mean density of the vertical column is negligible compared to the uncertainty in dh/dt. We use a mean density of 900 kg m$^{-3}$ which is commonly used for geodetic mass balance studies (e.g. Cogley, 2009; Haug et al., 2009). Scambos et al. (2014) use also a mean density of 900 kg m$^{-3}$ for converting volume change into mass for their mass balance analysis of glaciers on the northern Antarctic Peninsula over the period 2001 to 2010, so that these data can be directly compared with our estimates.

The radar backscatter coefficients (sigma-0) and structural patterns at the flux gates in the TerraSAR-X images correspond to typical features of glacier ice so that the density of ice can be used for computing the calving fluxes. The surface patterns and co-polarized sigma-0 values (-6 dB to -9 dB) in the transition zones from grounded ice to floating ice of SCAR inlet ice shelf are indicating blue ice. Frozen firn further upstream has typical sigma-0 values between -1 dB and -3 dB. The ice at the fronts of the calving glaciers is highly fractured so that corner effects cause higher sigma-0 values than those of blue ice, but the structural pattern is typical for exposed glacier ice. The Landsat image of 29 October 2016 shows reduced surface reflectivity for the lower terminus section of the calving glaciers, also an indication for glacier ice.

**Changes in manuscript:** We added relevant information in the revised manuscript (L278 to L283) and make reference to Supplement Section 3.2 for further details.

**Comment** 3. *The discussion is a bit disappointing in relation to the rest of the paper, and reads like an extended (repetitive) description of the results. I think one point which could be discussed more prominent in a broader context is the connection of mass loss and sea ice (fast ice) cover in the embayments. This could be an essential contribution to a better understanding of how ice shelves form, or the stability of existing ice shelves.*

**Response:** Thanks for pointing this out and the suggestions. We add more substance to the discussion by referring to studies on the buttressing effects of ice mélange for calving behaviour and discussing possible impacts of recent changes in climate conditions and atmospheric circulation.

In 2013 to 2016 the fjords and bays in front of the glaciers were covered by ice mélange, whereas the wider area of the Larsen A and B embayments was covered by sea ice of different age. To illustrate these features we include in the revised Supplementary Material a TerraSAR-X image covering the proglacial fjord of Crane, Jorum and Punchbowl glaciers (Figure S4). In the discussion we refer to publications on models and observations of the buttressing effect by ice mélange affecting the seasonal advance and retreat of calving fronts of Greenland outlet glaciers (Walter et al., 2012; Todd and Christofferson, 2014; Amundson et al., 2016). Major advancements on the role of proglacial ice mélange and persistent sea ice cover for ice shelf formation would require substantial modelling work which is beyond the scope of this paper.

**Changes in manuscript:** Discussion on the buttressing effect of ice mélange is added in L576 to L587. Discussion on atmospheric circulation and sea ice pattern is added in L598 to L610.

**Comment** 4: *Figures 5 and 6: Maybe it would be better to place the labels of the catchment areas outside of them and connect with an arrow. In the form they are now they are obscuring quite a large part of the smaller catchments. The label B12 is missing, but as it's the only one maybe that's intentional.*

**Changes in manuscript:** We modified Figures 5 and 6 as suggested and added the missing label.

[revised manuscript text omitted]